# Irradiance and cloud optical properties from solar photovoltaic systems

James Barry[1,2], Stefanie Meilinger[1], Klaus Pfeilsticker[2], Anna Herman-Czezuch[1], Nicola Kimiaie[1], Christopher Schirrmeister[1], Rone Yousif[1], Tina Buchmann[2], Johannes Grabenstein[2], Hartwig Deneke[3], Jonas Witthuhn[3], Claudia Emde[4], Felix Gödde[4], Bernhard Mayer[4], Leonhard Scheck[4,5], Marion Schroedter-Homscheidt[6], Philipp Hofbauer[7], and Matthias Struck[7]

[1]International Centre for Sustainable Development, Hochschule Bonn-Rhein-Sieg, Sankt Augustin
[2]Institute of Environmental Physics, University of Heidelberg
[3]Leibniz Institute for Tropospheric Research, Leipzig
[4]Meteorological Institute, Ludwig-Maximilians-Universität München
[5]Hans-Ertel Centre for Weather Research, Munich
[6]German Aerospace Center (DLR), Institute of Networked Energy Systems, Oldenburg
[7]egrid applications & consulting GmbH, Kempten

**Correspondence:** James Barry (james.barry@iup.uni-heidelberg.de)

**Abstract.**

Solar photovoltaic power output is modulated by atmospheric aerosols and clouds and thus contains valuable information on the optical properties of the atmosphere. As a ground-based data source with high spatiotemporal resolution it has great potential to complement other ground-based solar irradiance measurements as well as those of weather models and satellites, thus leading to an improved characterisation of global horizontal irradiance. In this work several algorithms are presented that can retrieve global tilted and horizontal irradiance and atmospheric optical properties from solar photovoltaic data and/or pyranometer measurements. The method is tested on data from two measurement campaigns that took place in the Allgäu region in Germany in autumn 2018 and summer 2019, and the results are compared with local pyranometer measurements as well as satellite and weather model data. Using power data measured at 1 Hz and averaged to 1-minute resolution along with a non-linear photovoltaic module temperature model, global horizontal irradiance is extracted with a mean bias error compared to concurrent pyranometer measurements of 5.79 W m$^{-2}$ (7.35 W m$^{-2}$) under clear (cloudy) skies, averaged over the two campaigns, whereas for the retrieval using coarser 15-minute power data with a linear temperature model the mean bias error is 5.88 W m$^{-2}$ and 41.87 W m$^{-2}$ under clear and cloudy skies, respectively.

During completely overcast periods the cloud optical depth is extracted from photovoltaic power using a lookup table method based on a one-dimensional radiative transfer simulation, and the results are compared to both satellite retrievals as well as data from the COSMO weather model. Potential applications of this approach for extracting cloud optical properties are discussed, as well as certain limitations, such as the representation of 3D radiative effects that occur under broken cloud conditions. In principle this method could provide an unprecedented amount of ground-based data on both irradiance and optical properties of the atmosphere, as long as the required photovoltaic power data are available and are properly pre-screened to remove unwanted artefacts in the signal. Possible solutions to this problem are discussed in the context of future work.

# 1 Introduction

An accurate determination of incoming solar radiation at the Earth's surface is important not only for both climate and weather research, but in future will also be vital for the stable operation of the electricity grid. In Germany alone there are 2.6 million photovoltaic (PV) systems installed, with a nominal power of 71 GWp (Holm, 2023), so that accurate forecasts of solar PV power generation are indeed becoming indispensable for cost-effective grid operation. In this context the proliferation of PV systems provides a unique opportunity to characterise global irradiance with unprecedented spatiotemporal resolution, which would lead to improvements in both weather and climate models. Solar panels can in this way be seen as a dense network of sensors for atmospheric optical properties. This new information could facilitate the development of highly resolved PV power forecasts, as well as play a role in improving climate models, in particular since the highly variable nature of cloud cover as well as uncertainties in cloud microphysics result in the greatest uncertainty in our understanding of the radiative forcing of the climate.

It has been shown by several authors [see for instance Urraca et al. (2018); Ohmura et al. (1998); Frank et al. (2018); Zubler et al. (2011)] that the estimates of global horizontal irradiance (GHI) from both the global ECMWF (ERA5) and regional (COSMO-REA6) numerical weather prediction (NWP) model reanalyses deviate from ground-based measurements. In Urraca et al. (2018), comparisons are made with pyranometer measurements from the Baseline Solar Radiation Network (BSRN) (Ohmura et al., 1998) as well as from a dense network of pyranometers operated by European meteorological services. In general the model reanalyses overestimate the irradiance under cloudy skies, which is largely due to an underestimation of cloud optical depth (COD). The mean positive bias of ERA5 daily mean irradiance is +4.05 W m$^{-2}$ (3.47%) over Europe and +4.54 W m$^{-2}$ (2.92%) worldwide. On the other hand, the regional COSMO-REA6 data set underestimates GHI on clear sky days, with a mean bias of -5.29 W m$^{-2}$ (-3.22%), which can be attributed to the use of an aerosol climatology with a too large aerosol optical depth (AOD), as discussed in Frank et al. (2018). Although the COSMO-D2 data uses a different aerosol scheme, these negative biases in the GHI are still present, especially in summer (Zubler et al., 2011). Satellite datasets perform a lot better, with data from the Solar surfAce RAdiation Heliosat (SARAH) showing a mean bias of only +0.86 W m$^{-2}$ in the daily mean GHI (compared to +4.22 W m$^{-2}$ from ERA5) over Europe (Urraca et al., 2018). Interestingly SARAH overestimates in most cases, with only some stations showing a negative bias related to snow detection. Overall the satellite measurements display a smaller absolute error than reanalysis products. The positive bias of the GHI from satellite retrievals is confirmed by Yang and Bright (2020): their comprehensive global evaluation of hourly satellite irradiance data reveals a mean bias error[1] of 4.67 W m$^{-2}$ for hourly SARAH-2 irradiance compared to the nine BSRN stations over Europe (excluding the Austrian station Sonnblick at 3100 m altitude), compared to 7.93 W m$^{-2}$ for the Copernicus Atmospheric Monitoring Service (CAMS) radiation data (cf. Section 3.3).

The idea of using PV systems as radiation sensors has been explored by several authors. In Engerer and Mills (2014), Killinger et al. (2016) and Marion and Smith (2017), methods are developed in order to use the output of one PV system to predict that of another, which is in essence done by inferring GHI from PV power measurements. In all three cases empirical

---

[1]Calculated using Table 3 in Yang and Bright (2020).

models for the decomposition of irradiance into direct and diffuse components are used, and system parameters such as ori-
entation and PV module efficiency are required inputs. Engerer and Mills (2014) achieve a mean bias error of 1.09% for the
PV power output under clear sky conditions, but the accuracy diminishes for partly cloudy skies, as expected; Killinger et al.
(2016) achieve a mean bias error between -3.9% and -9.8% for the GHI, depending on the empirical model used for irradiance
transposition; and Marion and Smith (2017) achieve a mean bias error for the GHI of within ±1.5% using south-facing PV
modules at 10°, 25° and 40° tilt angles. A similar approach is taken in Elsinga et al. (2017), in this case using a single diode PV
model and a different decomposition model. In Halilovic et al. (2019) the authors replaced the original iterative approach used
in Killinger et al. (2016) by an analytical method, to minimise computational cost, and achieve a mean bias error of between
0.1% and 2.1% for the resulting GHI, using data from silicon reference cell measurements at different tilt and azimuth angles.

In Nespoli and Medici (2017) a different method is introduced, in this case without the need for system-specific information
such as orientation or nominal power. A similar approach is taken in Saint-Drenan (2015); Saint-Drenan et al. (2015), where
system parameters are estimated by statistical methods. In addition, Scolari et al. (2018), Laudani et al. (2016), Carrasco et al.
(2014) and Abe et al. (2020) have also described the inference of solar irradiance from PV current and voltage measurements
using an equivalent circuit model. In this case greater accuracy is achievable, provided the module temperature is also measured.

This work builds upon the proof of concept study presented in Buchmann (2018) (for clear sky days only), however it
is unique in that empirical models for the separation of radiation components are avoided – rather an explicit simulation of
the diffuse radiance distribution is performed using libRadtran (Mayer and Kylling, 2005; Emde et al., 2016). Although this is
computationally more intensive it has several advantages over the usual approach [see for instance Perez et al. (1992)]: by using
a state-of-the-art radiative transfer code one can more accurately model the clear sky irradiance, especially for larger solar zenith
angles, and one can explicitly take into account information on aerosol load or ground albedo from freely available datasets.
In addition it is possible to include information on the state of the atmosphere from weather models, which is particularly
relevant in including the effects of precipitable water on incoming irradiance. The radiative transfer solvers DISORT (Stamnes
et al., 1988; Buras et al., 2011) and MYSTIC (Mayer, 2009) are used for forward model calibration as well as for inferring
atmospheric optical properties and GHI from ground-based irradiance measurements and/or PV power data.

In order for a PV-based determination of solar irradiance to viably complement the global coverage of state-of-the-art satellite
measurements, a mean bias error of the order of 5 W m$^{-2}$ would be desirable (see the discussion on CAMS and other satellite-
based products above). This level of accuracy also corresponds to the target accuracy for global radiation measurements from
the BSRN (McArthur, 2005). However, even if this is not achieved, ground-based irradiance measurements and/or retrievals can
be seen as complementary since they have the added advantage of superior spatiotemporal resolution. The first step to achieve
this is to accurately model the generated power as a function of system-specific parameters, such as the array's elevation and
azimuth angle, conversion efficiency and temperature dependence, and then extract those parameters from measured power data
using a fitting procedure. In order to remove any biases related to atmospheric conditions, it makes sense to first calibrate the
systems under clear skies. Once this has been done to sufficient accuracy one can use measured PV power to infer atmospheric
optical parameters such as aerosol or cloud optical depth under different sky conditions, enabling the inference of GHI as well
as in some cases of direct and diffuse irradiance components.

The more parameters used to model the PV power, the greater the uncertainty in the retrieved irradiance. For this reason it is of course desirable to obtain as much a priori metadata about the PV systems as possible, such as datasheet parameters and array orientation. However, this information is not always readily available, especially when considering a large amount of PV systems over a wide area. In that sense, there will always be a trade-off between quantity and quality of the data, which then plays itself out in the accuracy of the retrieved irradiance. The advantage of PV systems or any ground-based devices is that one can achieve a much higher spatiotemporal resolution compared to satellite data or weather models, which thus allows one to study high-frequency fluctuations of global irradiance.

In the European context, irradiance variability is dominated by the optical properties of clouds and less by those of aerosols. Ground-based COD retrievals using broadband measurements from pyranometers have been carried out in several studies (see for example Leontyeva and Stamnes (1994); Boers (1997); Boers et al. (1999); Deneke (2002)). Indeed, the transmission of irradiance through a cloud is most sensitive to its optical depth, and less sensitive to droplet radius, single scattering albedo or asymmetry factor (Leontyeva and Stamnes, 1994). In most previous studies the clouds are assumed to be horizontally homogeneous in a plane-parallel atmosphere with 1D radiative transfer, which leads to a bias in the extraction of cloud optical properties, in particular under broken cloud conditions. By neglecting 3D effects, the horizontal transport of photons is not considered, which however plays an important role in real life situations. These 3D effects can for example lead to an enhancement of solar irradiance (Schade et al., 2007), so that the GHI exceeds the clear sky irradiance due to reflected light from the edges of clouds. The inherent four dimensional variability of clouds also complicates the comparison of ground-based and satellite retrievals of cloud properties, since one compares the time average of a point measurement with a spatially averaged quantity.

The goal of this work is to demonstrate that PV systems can indeed be used as ground-based sensors for both GHI as well as to infer the optical properties of the atmosphere, in particular the COD. First results are presented from two measurement campaigns carried out in autumn 2018 and summer 2019 in the Allgäu region in southern Germany, as part of the BMWi-funded project MetPVNet (Meilinger et al., 2021). In Section 2 the forward model and its calibration are discussed, and the inversion methods are outlined in detail. Section 3 provides a detailed description of the data from the measurement campaigns. The results are presented in Section 4, with a focus on both tilted and horizontal irradiance as well as cloud optical depth under overcast skies, and a summary and conclusions are given in Section 5. Further details of the PV modelling aspects and radiative transfer simulation are found in the Appendix.

## 2   Photovoltaic power model: calibration and inversion

In order to infer local atmospheric optical properties from measured PV data, accurate modelling of both atmospheric radiative transfer as well as PV power generation is required. In this section both the PV model as well as the libRadtran radiative transfer model is described, along with the calibration and inversion procedure.

## 2.1 Forward model: from atmospheric properties to photovoltaic power

The power generated by a solar PV module depends primarily on incoming short wave solar irradiance and module temperature, both of which depend on atmospheric conditions. Once this dependence is properly described in a model, PV power and/or current measurements can be used to infer the irradiance in the plane of the array, taking into account the geometry of the system, i.e. the elevation and azimuth angle of the solar panels. After extracting this "global tilted irradiance" (GTI) from PV data, one can go on to infer atmospheric optical properties such as cloud optical depth and global horizontal irradiance, by

further inverting the radiative transfer model.

    The most physically correct method of modelling the power output of a PV plant is with an equivalent circuit model that captures the properties of semiconductors, such as the two-diode model (see for instance Mertens (2014)). In this way the temperature dependence of current and voltage is explicitly defined according to the Shockley equation (Shockley, 1949). A drawback of such models is their computational complexity and reliance on parameters found on module datasheets, which are

in the most general case not always available. There are however several parameterised models in the literature that attempt to reduce the power generation equation to a simple relation between incoming plane-of-array irradiance, module area and temperature-dependent efficiency, with the latter described as a function of ambient conditions. Several such models exist (see Skoplaki and Palyvos (2009) for a review), with some of the most popular being that of the "PV Performance Modeling Collaborative" from Sandia National Laboratories (King et al., 2004, 2007) or the Huld model used in the online PVGIS tool

(Huld et al., 2011). Since the goal here is an inversion, the choice of model depends on the availability of measured data: in this work and in the context of the AC power data from the MetPVNet campaign, a simplified parametric power model is employed. The model is described here briefly, and more details are given in Appendix A.

    In order to correctly capture the effects of the variable solar spectrum one also needs to take into account the spectral response of the PV technology in question (Alonso-Abella et al., 2014), which in the case of an equivalent circuit model can

then be included in the calculation of the photocurrent [see for instance the libRadtran-based spectral PV model described in Herman-Czezuch et al. (2022)]. In the case of parametrised PV power models, this so-called "spectral mismatch", i.e., the difference between the entire spectrum of incoming radiation and the range utilised by a certain PV module, is usually simply absorbed into the PV model parameters, leading to a site-specific bias that may not take into account variations in the spectrum from local atmospheric conditions. By using libRadtran for calibration and inversion along with information on the state of

the atmosphere from weather models one can take these variations into account in the radiative transfer (RT) simulation and subsequent inversion, as discussed in 2.2 below. In particular the water vapour column and aerosol optical depth at each site need to be taken into account (see Section A3 in the Appendix).

    It can be shown using the diode model [see for instance Sauer (1994); Abe et al. (2020)] that the maximum power point (MPP) current generated by a PV module is linearly dependent on the incident irradiance, and only very weakly dependent

on temperature. However, the dependence of MPP voltage on temperature (which itself is a function of irradiance) is an order of magnitude greater (roughly -0.4 %/K), so that this simple linear relationship breaks down when considering the PV power. In this work a parameterised power model is used [see Buchmann (2018), Skoplaki and Palyvos (2009), as well as Dows and

Gough (1995)], with AC PV power described as[2]

$$P_{\mathrm{AC,mod}} \simeq G_{\mathrm{tot,PV},\tau}^{\angle} \left( b_1 + b_2\, G_{\mathrm{tot,SW},\tau}^{\angle} + b_3\, T_{\mathrm{ambient}} + b_4\, v_{\mathrm{wind}} + b_5\, T_{\mathrm{sky}} \right), \tag{1}$$

in the case of the linear temperature model defined in Eq. (A3) (TamizhMani et al., 2003), or as

$$P_{\mathrm{AC,mod}} \simeq G_{\mathrm{tot,PV},\tau}^{\angle} \left( b_1' + \frac{G_{\mathrm{tot,SW},\tau}^{\angle}}{b_2' + b_4'\, v_{\mathrm{wind}}} + b_3'\, T_{\mathrm{ambient}} + b_5'\, T_{\mathrm{sky}} \right), \tag{2}$$

in the case of the non-linear temperature model defined in Eq. (A4) (Faiman, 2008; Barry et al., 2020). This means that the modelled AC power $P_{\mathrm{AC,mod}}$ is a non-linear function of plane-of-array irradiance $G_{\mathrm{tot,PV},\tau}^{\angle}$, together with the effects of ambient temperature $T_{\mathrm{ambient}}$, wind speed $v_{\mathrm{wind}}$ and sky temperature $T_{\mathrm{sky}}$ that influence module temperature and thus
efficiency. Note that the subscript "PV" for the tilted irradiance $G_{\mathrm{tot,PV},\tau}^{\angle}$ refers to the fact that only the relevant wavelength (in this case 300 nm to about 1200 nm for silicon PV modules) is considered, and the subscript "$\tau$" indicates that transmission through the glass surface of the PV panels has been taken into account with an optical model. Further details of the model employed here are given in Sections A1 and A2 in the Appendix, and the refractive index $n$ of the glass covering is one of the parameters varied in the optimisation procedure. The subscript "SW" refers to all incoming shortwave photons – the
dependence of the spectral mismatch between the PV and SW irradiance bands on atmospheric water vapour and other factors is discussed in Section 2.3.

The parameters $b_i(b_i')$, $(i = 1 \ldots 5)$ in Eqs. (1) and (2) depend on nominal power, efficiency, the temperature coefficient for power as well as the temperature model parameters, and are discussed in more detail in Appendix A, which includes a list of all parameters in Table 2. In practice the module temperature can either be measured or modelled, depending on the availability of
measurements and/or meteorological data. Within the PV power models described above, the PV module temperature is a static quantity, i.e., the heat capacity ($C$) of the PV system is not taken into account. However, when dealing with high-frequency measurements of PV power it is necessary to employ a dynamic temperature model, as discussed in Barry et al. (2020). The characteristic time constant ($= C/J$, with $J$ the net thermal energy flux of the PV modules) of typically 10 minutes means that the large fluctuations in irradiance do not translate directly to module temperature variations, i.e. the temperature response is
smoothed out. For simplicity the dynamic temperature model is not included in the present study, since most of the systems had power data collected in 15 minute resolution.

## 2.2 Model calibration under clear sky conditions

In order to infer the irradiance in the plane-of-array (GTI) from measured PV power or current, the PV model parameters need to be determined, either from datasheets or with a forward model calibration. This is accomplished using data under clear
sky conditions, together with an accurate simulation of the irradiance, followed by a multi-parameter optimisation to find the parameter values. This section describes the technical details of the clear sky simulation and the relevant atmospheric input parameters used. Figure 1 displays this procedure graphically, and further explanations are given in the following sections.

---

[2]The inverter efficiency is included in the parameter $s$, see Section A in the Appendix.

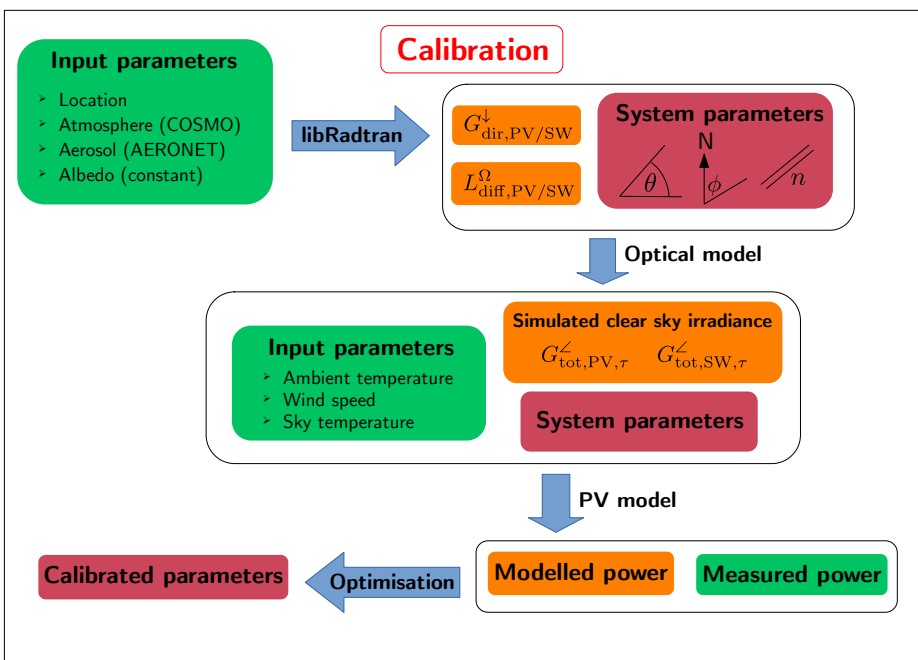

**Figure 1.** Schematic diagram showing the different steps of the calibration procedure, with input data sources in green, model steps/algorithms in blue, simulated parameters in orange and system parameters (cf. Table 2) in dark red. Note that in this case only clear sky days or time periods are considered.

**Table 1.** Model parameters for libRadtran simulation of clear sky days, including information on their source.

| Parameter | Symbol | Source |
|---|---|---|
| Latitude, longitude, altitude, time | $\varphi, \vartheta, z, t$ | Set by PV data |
| Solar zenith angle, solar azimuth angle | $\theta_0, \phi_0$ | Calculated with `PyEphem` (Rhodes, 2022) |
| Temperature profile | $T(z)$ | COSMO (Baldauf et al., 2011) |
| Pressure profile | $p(z)$ | COSMO |
| Water vapour | $[H_2O](z)$ | COSMO |
| Ozone | $[O_3](z)$ | US standard atmosphere |
| Albedo | $\rho(\lambda)$ | Constant (0.2) |
| Ångström turbidity coefficient | $\tau_{a,1}$ | AERONET (Holben et al., 1998) |
| Ångström exponent | $\alpha$ | AERONET |
| Other aerosol optical properties | - | OPAC "continental average" (Hess et al., 1998) |

### 2.2.1 Radiative transfer simulation with libRadtran

The clear sky simulation of tilted irradiance $G_{\text{tot,PV},\tau}^{\angle}$ is performed with the freely available libRadtran software package (Mayer and Kylling, 2005; Emde et al., 2016), with the input parameters shown in Table 1 and the wavelength range 300 nm to 1200 nm for silicon PV applications. The corresponding broadband simulation ($G_{\text{tot,SW},\tau}^{\angle}$) is also performed, as an input to the temperature model. Spectral integration is carried out using the Kato parameterisation, in order to simplify the effects of water vapour absorption by using the so-called correlated-k approximation (Kato et al., 1999). The DISORT solver allows for an explicit calculation of the diffuse radiance distribution on a predefined lattice of elevation and azimuth angles, and the pseudospherical approximation is employed, so that only radiative transfer calculations at solar zenith angles (SZA) of up to 80 degrees can be reliably performed. The Python package `PyEphem` (Rhodes, 2022) is used to accurately determine the sun position for the corresponding latitude, longitude, time coordinates. COSMO model data (see Section 3.2) are interpolated by the package `cosmomystic` (see software supplement) in both time and space in order to create atmosphere profile files suitable as input for libRadtran, in 15 minute resolution. In this way variations in water vapour and other atmospheric trace gases are taken into account, and the atmospheric layers are cut off at the appropriate altitude of each site. Concurrent measurements by an AErosol RObotic NETwork (AERONET) sun photometer (Holben et al., 1998) are used to extract the Ångström exponent $\alpha$ and turbidity coefficient $\tau_{a,1}$ with the `aeronetmystic` (see software supplement) software package. Other aerosol optical properties such as the single scattering albedo and asymmetry factor are taken from the Optical Properties of Aerosols and Clouds (OPAC) species library with the option "continental average" (see Table 3 in Hess et al. (1998)).

In order to speed up the simulation the code is parallelised to run on multiple processors: the simulaton times are divided up into batches and libRadtran is then called multiple times as a subprocess from Python. In this way the clear sky simulation takes approximately 1 second per time step (8 seconds on an 8 core machine), using a diffuse radiance field of $5°$ resolution in both elevation and azimuth angle, atmosphere files modified from COSMO data and modified aerosol inputs from AERONET and OPAC.

### 2.2.2 Non-linear optimisation for PV system parameters

The simplified parametric model described above can be written as

$$P_{\text{AC,mod}} \equiv \boldsymbol{y} = \boldsymbol{F}(\boldsymbol{x}, \boldsymbol{h}), \tag{3}$$

for the forward model $\boldsymbol{F}$ described by Eq. (A1) and state space defined by (cf. Table 2)

$$\boldsymbol{x} \equiv (\theta, \phi, n, s, \zeta, u_i), \tag{4}$$

so that the calibration procedure is effectively a non-linear, multi-parameter optimisation problem with eight (for the non-linear temperature model)[3] or nine (for the linear temperature model) unknowns. As shown in Table 3, the parameter space $\boldsymbol{h}$ in Eq. (3) contains the irradiance proxy from the libRadtran simulation as well as temperature and wind speed data from either the COSMO model or measurements, which are interpolated to 15 minute resolution. In addition the measured sky temperature

---

[3]In the non-linear Faiman model there are less temperature parameters as the ambient and sky temperatures are not independent, as is the case in the linear model, cf. Eqs. (A3) and (A4).

**Table 2.** List of PV model parameters in $\boldsymbol{x}$. In the calibration procedure, those parameters in $\boldsymbol{x}$ known to a certain degree (from datasheets or other sources) of accuracy are fixed, whereas all others are varied.

| Parameter ($\boldsymbol{x}$) | Symbol | Source (if available) |
|---|---|---|
| Tilt angle | $\theta$ | laser scanning and/or theodolite |
| Azimuth angle | $\phi$ | laser scanning and/or theodolite |
| Glass refractive index | $n$ | optimisation |
| Scaling factor | $s$ | optimisation |
| Temperature coefficient | $\zeta$ | datasheet and/or optimisation |
| Temperature model parameters | $u_i\ (i=0,1,2,3)$ | optimisation and/or model (Barry et al., 2020) |

**Table 3.** List of additional inputs in $\boldsymbol{h}$ used for calibration on clear sky days. The subscripts "PV" and "SW" refer to the different wavelength bands used for integration, see the discussion in Section 2.2.1.

| Parameter ($\boldsymbol{h}$) | Symbol | Source |
|---|---|---|
| Direct tilted irradiance | $G_{\mathrm{dir,PV(SW)}}^{\angle}$ | libRadtran simulation (see Table 1) |
| Diffuse tilted irradiance | $G_{\mathrm{diff,PV(SW)}}^{\angle}$ | libRadtran simulation (see Table 1) |
| 2m ambient temperature | $T_{\mathrm{ambient}}$ | COSMO and/or measured |
| Wind speed at 10m | $v_{\mathrm{wind}}$ | COSMO and/or measured |
| Longwave downward welling irradiance | $G_{\mathrm{LW}}^{\downarrow}$ | measured |

(see Section 3.1) is used. This inversion problem can be solved with the methods detailed in Rodgers (2000). In this case the

215 Levenberg-Marquardt algorithm is used, with the Jacobian matrix calculated explicitly at each iteration.

If one varies all parameters in $\boldsymbol{x}$ it quickly becomes apparent that there are not enough degrees of freedom in the signal to uniquely determine a solution with the Bayesian formalism, since several parameters are highly correlated with each other, for instance the inclination angle $\theta$ with the scaling factor $s$, or the orientation angle $\phi$ with the temperature model parameters or the coefficient $\zeta$ (cf. the discussion in Section 4.1). It is thus expedient to extract the temperature model parameters separately

using measured module temperature for different PV system configurations (if available) and then fix those parameters in the optimisation procedure. In Barry et al. (2020) a dynamic model was developed by fitting the measured and modelled module temperature using three different PV systems with different mountings. These results (for the static model case) are used in the overall optimisation, where appropriate.

The calibration algorithm is designed to allow certain parameters to be fixed if they are known, whereas unknown parameters

are varied with a given a priori error, which in turn affects the parameter retrieval error and thus propagates into the uncertainty in the inferred irradiance.

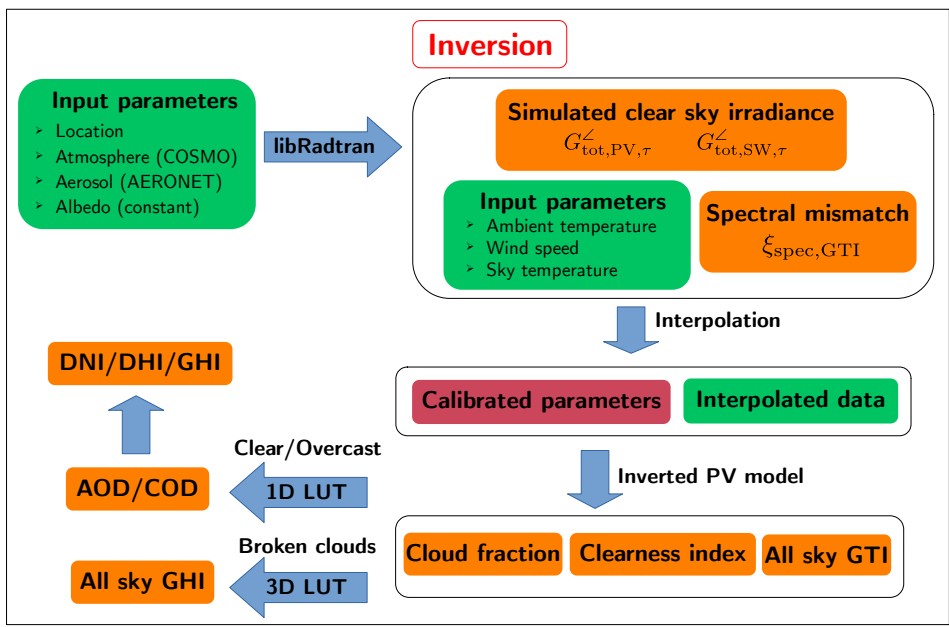

**Figure 2.** Schematic diagram showing the different steps of the inversion procedure, with input data sources in green, model steps/algorithms in blue, simulated/retrieved parameters in orange and system parameters in dark red. Note that in this case all atmospheric conditions (all sky) are considered.

## 2.3 Model inversion under all sky conditions

The calibrated PV systems can now be used as sensors to extract information about the state of the atmosphere. This section describes the different methods used to infer both irradiance and atmospheric optical properties from PV power data, as sum-
230 marised in the schematic diagram in Figure 2. The method employed depends on the prevailing weather conditions, specifically on the amount (and type) of cloud cover. In a nutshell, using a 1D DISORT-based method one can use GTI to extract AOD or COD during clear or completely overcast periods, respectively, which thus allows the determination of the direct and diffuse irradiance components. Under broken cloud conditions, a 3D MYSTIC-based method allows one to determine the GHI from GTI directly. The DISORT and MYSTIC lookup tables (LUTs) are provided in an open data repository.

### 2.3.1 Global tilted irradiance from PV model inversion

Once the PV system has been calibrated under clear sky conditions, the system parameters can be fixed and the measured PV power can be used to infer the global tilted irradiance (GTI, also denoted as $G_{\text{tot,SW}}^{\angle}$) under all sky conditions. The temperature model makes use of broadband irradiance [see Eqs. (1) and (2)], whereas the PV power model uses only the relevant spectral

range of silicon PV modules (300 nm-1200 nm), so that the spectral mismatch between the light converted to electricity ($G^{\angle}_{\text{tot,PV},\tau}$) and the entire shortwave spectrum needs to be taken into account when inverting the model chain.

The ratio of PV-relevant ($G^{\angle}_{\text{tot,PV}}$) to broadband ($G^{\angle}_{\text{tot,SW}}$) tilted irradiance is a function of system geometry, time of day as well as local atmospheric conditions, with the largest contributing factor being the precipitable water in the atmosphere. In order to take this into account the libRadtran clear sky irradiance simulations (see Section 2.2.1) are used to characterise the ratio

$$\xi_{\text{spec,GTI}} \equiv \frac{G^{\angle}_{\text{tot,PV},\tau}}{G^{\angle}_{\text{tot,SW},\tau}} = f(\Theta, [\text{H}_2\text{O}], AOD), \tag{5}$$

as a function of incident angle $\Theta$, precipitable water and aerosol optical depth, for each station and measurement campaign. In this way the available information on water vapour column and aerosol extinction from the COSMO model and AERONET can be taken into account in the PV model inversion. Details are given in Section A3 in the Appendix. The fitting function could in principle be extended to include ozone column abundance, which is however not included here, since this information is not available from the COSMO model data. Note that although this method does not take into account the effect of clouds on the spectral mismatch it is a good first approximation, which will be improved upon in future work (see also Rivera Aguilar and Reise (2020) for an alternative method). In addition one could modify this algorithm to include operational satellite retrievals of atmospheric parameters such as ozone concentration, if required.

Once the spectral mismatch factor $\xi_{\text{spec,GTI}}$ has been calculated, the next step is to extract the plane-of-the-array irradiance from PV power, which in the case of the models given in Eqs. (1) and (2) is simply the solution to the quadratic equations in $G^{\angle}_{\text{tot,SW},\tau}$, i.e.,

$$\xi_{\text{spec,GTI}}\, b_2 \left(G^{\angle}_{\text{tot,SW},\tau}\right)^2 + (b_1 + b_3\, T_{\text{ambient}} + b_4\, v_{\text{wind}} + b_5\, T_{\text{sky}})\, \xi_{\text{spec,GTI}}\, G^{\angle}_{\text{tot,SW},\tau} - P_{\text{AC,meas}} = 0\,, \tag{6}$$

for the linear temperature model and

$$\frac{\xi_{\text{spec,GTI}}}{(b_2 + b_4\, v_{\text{wind}})} \left(G^{\angle}_{\text{tot,SW},\tau}\right)^2 + (b_1 + b_3\, T_{\text{ambient}} + b_5\, T_{\text{sky}})\, \xi_{\text{spec,GTI}}\, G^{\angle}_{\text{tot,SW},\tau} - P_{\text{AC,meas}} = 0\,, \tag{7}$$

for the non-linear temperature model. These equations can be solved with the quadratic formula, using the calibrated parameters $b_{1,2,3,4,5}$ ($b'_{1,2,3,4,5}$) as defined in Eq. (A5) (Eq. (A6)) for the linear (non-linear) temperature model, the available data for $T_{\text{ambient}}$, $v_{\text{wind}}$ and $T_{\text{sky}}$ as well as the spectral mismatch factor for GTI defined in Eq. (5). Note that the inverted $G^{\angle}_{\text{tot,SW},\tau}$ is the irradiance impinging upon the PV module under the glass covering, so that the optical model has not yet been inverted. In order to compare this quantity with pyranometers, the transmission of light through the glass $\tau_{\text{PV,rel}}(\Theta)$ [also known as "incidence angle modifier", see Eq. (A9)] must be taken into account, so that the final GTI is given by [see also Eq. (A15)]

$$G^{\angle}_{\text{tot,SW}} = \frac{G^{\angle}_{\text{tot,SW},\tau}}{\tau_{\text{PV,rel,eff}}}\,. \tag{8}$$

For the direct extraction of GTI an empirical formulation is used to find the effective incidence angle for the diffuse component, whereas for the inversion onto optical properties the refractive index is explicitly taken into account within the radiative transfer simulation. More details are given in Section A2 in the Appendix.

### 2.3.2 Clearness index and irradiance variability classification

Using the global tilted irradiance extracted from measured PV power data, different methods are used in order to extract atmospheric optical properties and global horizontal irradiance, depending on the prevailing weather conditions. By combining the inverted tilted irradiance with the corresponding clear sky curve one can calculate a clearness index

$$k_i(t) = \frac{G^{\angle}_{\text{tot,SW},\tau,\text{inv}}(t)}{G^{\angle}_{\text{tot,SW},\tau,\text{clear}}(t)} \tag{9}$$

for each time step, allowing the data to be separated into clear, overcast and broken cloud time periods. The clearness index is then used to estimate the cloud fraction, which is discussed in more detail in Section 2.3.3.

On clear days (or during clear time periods) the aerosol optical depth (AOD) can be inferred, whereas under cloudy conditions the cloud optical depth (COD) can be found, depending on the degree of cloud cover. In this work the extraction of COD using a DISORT-based LUT under completely overcast skies is examined in more detail in Section 2.3.3. An in-depth analysis of aerosol optical properties will be carried out in future work. For broken cloud conditions, a MYSTIC-based LUT is used to infer the global horizontal irradiance from tilted irradiance measurements, as discussed in Section 2.3.5 [see Chapter 9 of Meilinger et al. (2021)].

### 2.3.3 Cloud optical depth with DISORT lookup table

Cloud optical properties are functions of microphysical properties such as droplet size distribution, droplet number concentration as well thermodynamic phase. For water clouds, the absorption and scattering of solar irradiance can be efficiently characterised (Hu and Stamnes, 1993) by the effective radius $r_{\text{eff}}$ and cloud liquid water content (LWC), which can be related to cloud optical depth (COD, $\tau_c$) through

$$\tau_c = \frac{3\,\text{LWP}}{2\,r_{\text{eff}}\,\rho_{\text{H}_2\text{O}}}, \tag{10}$$

where the liquid water path (LWP) is the integral of the LWC across the height of the cloud. The derivation of this equation [see for instance Petty (2006)] assumes large Mie extinction, which is justified since clouds appear to be (mostly) white in the solar spectrum. Although both $\tau_c$ and $r_{\text{eff}}$ can be accurately retrieved from spectral measurements of reflected radiation, the transmission of light through clouds is mostly sensitive to the cloud optical depth. This is due to the fact that changes in transmission due to variations in single scattering albedo and asymmetry factor (which depend on $r_{\text{eff}}$) are small compared to those due to changes in optical depth (Leontyeva and Stamnes, 1994). For illustration, in the two stream approximation for conservative scattering (no absorption), the transmittance $T$ can be shown to be [see for instance Petty (2006)]

$$T = \frac{1}{1 + (1 - g)\,\tau_c}, \tag{11}$$

where $g$ is the asymmetry factor. For liquid water clouds, scattering is mostly in the forward direction, with $g \simeq 0.85$, whereas for ice clouds $g \simeq 0.7$. In both cases the variations in $g$ are small, so that $\tau_c$ is the primary factor influencing $T$. The hyperbolic dependency of $T$ on $\tau_c$ means that the transmission curve is rather steep at small optical depths, but flattens out for COD $\gtrsim 15$.

**Table 4.** Cloud parameters for the DISORT simulation of a continental stratus cloud (Hess et al. (1998)).

| Cloud parameter | Value |
|---|---|
| Liquid water content (LWC) | $0.28 \text{ g m}^{-3}$ |
| Effective radius ($r_{\text{eff}}$) | $7.33 \ \mu\text{m}$ |
| Cloud height ($h$) | $1-2 \text{ km}$ |

This has implications for the accuracy of ground-based retrievals, as discussed in detail in Section 4.4. It must also be noted that in the algorithm described in Section 2.3.1, spectral variations in cloud optical properties are not taken into account. In practice this means that variations in the single scattering albedo at higher wavelengths around 1 $\mu$m (silicon PV modules are still sensitive to wavelengths up to 1.2 $\mu$m) may be unaccounted for.

In this work a lookup table for the optical depth of a typical stratus cloud is constructed using DISORT in 15 minute time
intervals, under the assumption of a pseudospherical or plane-parallel atmosphere with horizontally homogeneous liquid water clouds and a completely cloudy sky. This means that 3D effects are not taken into account and the results need to be interpreted with care, especially in situations with broken clouds. In addition, different cloud types such as thicker cirrus clouds, mixed phase or multi-layer clouds are not properly represented by the LUT. Due to the pseudospherical approximation only SZA up to 75° are considered (for SZA above 75° with cloud cover the pseudospherical DISORT solver is unstable). The cloud
parameters in Table 4 are input into libRadtran and the COD at 550 nm ($\tau_{c,550\text{nm}}$) is varied on a 16 step logarithmic scale between COD = 0.5 and COD = 150, using the default "hu" parameterisation (Hu and Stamnes, 1993) and 16 streams. Note that the COD LUT also implicitly contains aerosol information as an input, since here the aerosol properties are fixed using the OPAC database (Hess et al., 1998), and the Ångström parameters from AERONET are used.

As described for the clear sky simulation in Section 2.2 the direct irradiance and diffuse radiance field are calculated with
libRadtran, the latter in this case with a coarser resolution of 10 degrees in both azimuth and elevation angles.[4] The LUT is then used to find the COD by first calculating the plane-of-array irradiance for the corresponding PV system or pyranometer orientation (cf. Section 2.3.1) and then interpolating the COD in time to match the resolution of the measured data. For this purpose the original 1 Hz pyranometer and PV data are smoothed to 1-minute resolution, whereas the low frequency PV data is kept at 15 minute resolution (see Section 3.1).

In order to determine the exact time points at which a cloud is above the sensor, the cloud fraction is determined by creating a mask based on the clearness index in Eq. (9) using a threshold of 0.8 and overshoot limit of 1.1, i.e.,

$$\text{cf}(t) = \begin{cases} 1 & \text{if } k_i(t) \leq 0.8 \\ 0 & \text{if } 0.8 < k_i(t) \leq 1.1 \\ \text{nan} & \text{if } k_i(t) > 1.1 \end{cases} \quad (12)$$

---

[4]Note that for more accurate radiance calculations one could use the "mie" option in libRadtran, which uses pre-calculated tables for Mie scattering and is however computationally more expensive.

This binary cloud mask (clear = 0, cloudy = 1) is then also smoothed with a moving average function over 60 minutes in order to create an estimate of the cloud fraction ($\langle cf \rangle_{60}$). Varying the thresholds in Eq. (12) shows that the cloud fraction computed in this way depends less on the exact threshold used, but more on the window size chosen for the moving average. Indeed, comparison with concurrent cloud camera retrievals shows that 60 minutes is a reasonable averaging time to use, when averaging a cloud mask created with data at 1-minute resolution. However, the algorithm is limited by the viewing angle of the respective PV system or pyranometer, so it can be inaccurate when there are many clouds on the horizon, for instance.

The COD is then only extracted for data points for which cf$(t) = 1$, i.e. by finding the values of $\tau_{c,550nm}$ for which

$$G^{\angle}_{tot,SW,meas/inv} = G^{\angle}_{dir,SW,cloudy}(\tau_{c,550nm}) + G^{\angle}_{diff,SW,cloudy}(\tau_{c,550nm}), \tag{13}$$

for all points under a cloud, where "meas" or "inv" refer to measured or inverted GTI from pyranometers or PV systems, respectively. The corresponding direct and diffuse components can then also be extracted from the LUT, although in this case the direct irradiance is basically zero (beneath a cloud).

As mentioned above, this approach is limited by the fact that a 1D radiative transfer solver such as DISORT cannot take into account horizontal transport of photons, so that 3D effects such as radiative enhancement under broken cloud conditions [see for instance Schade et al. (2007)] are not taken into account. For this reason only situations with overcast conditions will be considered when applying this method. In situations with low overall cloud cover, the COD is not the main determinant of the total irradiance received by the sensor or PV system, but rather the cloud fraction and/or the AOD. To this end a complementary approach using a MYSTIC-based LUT (see Section 2.3.5) is used, in order translate measured tilted irradiance to horizontal irradiance under broken cloud conditions.

### 2.3.4 Aerosol optical depth with DISORT lookup table

As mentioned above, in this work the extraction of the AOD will not be discussed in detail. However the procedure will be briefly described here, since this is used as an alternative method for determining the GHI from tilted irradiance measurements. An AOD-GTI lookup table can be created in a similar way to the COD LUT described in Section 2.3.3, where in this case the AOD at 500nm is varied on a logarithmic scale in 16 steps between AOD = 0.01 and AOD = 1. In addition, the aerosol properties are fixed to the so-called "continental average" scheme from the OPAC database (Hess et al., 1998), and for the inversion procedure the AERONET-based Ångström parameters are not used as input. In this context it must be noted that the typical dust event reaching Europe does not have such a high AOD, but is rather characterised by small values of the Ångström exponent less than 1, indicating the presence of coarser dust particles. For example, one study of the climatology of dust events found a mean AOD of 0.155, 0.32 and 0.122 for dust plumes in southern, central and northern Europe (Mandija et al., 2018).

Using the AOD-GTI lookup table, the AOD can be extracted on clear sky days, and from this also the direct and diffuse irradiance as well as the global horizontal irradiance. In this way the AOD is used as an intermediate step for the reverse transposition of GTI to GHI. In Germany and especially in the Allgäu region the AOD is usually very small (during the measurement campaigns it did not exceed 0.5 at 550nm), so that any errors in the calibration procedure lead to large relative biases in the AOD. This then leads to biases in the direct and diffuse components, but since there are very few absorbing

**Table 5.** Limits on the input parameters for the MYSTIC LUT.

| Input parameter | Limits |
| --- | --- |
| SZA ($\theta_0$) | $[20°, 60°]$ |
| Tilt angle ($\theta$) | $[0°, 50°]$ |
| Relative azimuth ($|\phi - \phi_0|$) | $[0°, 90°]$ |
| Cloud fraction | $[0.13, 0.82]$ |

aerosols, these errors have opposite signs and largely cancel out in the determination of the GHI. In Section 4, the GHI extracted via both COD and AOD under different conditions is compared to that measured by pyranometers and satellites, as well as the GHI predicted by the COSMO weather model. However the inferred AOD itself is not examined in detail.

### 2.3.5   From tilted to horizontal irradiance with MYSTIC lookup table

In order to extract the global horizontal irradiance from the tilted irradiance (from pyranometers or PV systems) a MYSTIC-based LUT for the GHI was developed using LES cloud fields (Črnivec and Mayer, 2019), taking into account various factors such as albedo, water vapour, sensor geometry and cloud fraction. Detailed 3D radiative transfer simulations were carried out and the most important factors turn out to be simply the sensor and sun geometry as well as the cloud fraction. A detailed description of the MYSTIC LUT is given in Chapter 9, Section 9.1.5 of Meilinger et al. (2021).

Table 5 shows the limits of applicability of the MYSTIC LUT, for which there are three major reasons. Firstly, despite several optimisations like the reduction of the number of photons used for the Monte Carlo simulations, the computational demand for calculating the LUT is high. For this reason, $20°$ is chosen as the lower limit for the SZA, since in the latitudes under investigation no smaller values occur. Similar considerations apply to the tilt angle of PV panels – in Allgäu, Germany one rarely encounters title angles larger than $50°$. The second limiting factor relates to the derivation of a cloud mask and cloud

fraction from the radiation measurements [see Eq. (12)]. Firstly, this is only possible when there is direct line of sight between the sun and the module or sensor, which limits the relative azimuth angle between the sun and the PV panel. Secondly, the derivation of cloud fraction from temporally resolved radiation measurements becomes imprecise at large SZAs for geometrical reasons, so that the upper limit of the SZA is set to $60°$. Finally, the cloud fraction limits are determined by two factors: firstly, the LUT model was developed and tested for partly cloudy situations. The special cases of 0% and 100% cloud fraction are

considered separately with the DISORT-based LUTs, as other parameters like AOD and COD become relevant here. Secondly, the exact cloud fraction limits (0.13 and 0.82) given in Table 5 are constrained by the available cloud scenes from LES simulations.

The measured or inverted tilted irradiance, together with the average cloud fraction over the last hour [as described above, cf. Eq (12)] is fed into the MYSTIC LUT, along with the sensor and sun geometry. In this way the GHI can be extracted from

the GTI under broken cloud conditions. This method can however not be used to determine the optical depth, nor can the direct and diffuse irradiance components be separated from each other, since the fit was created using the GTI and GHI.

**Table 6.** Dates of clear sky days during the measurement campaigns in autumn 2018 and summer 2019.

| | | | | | | |
|---|---|---|---|---|---|---|
| 1st campaign | 12 Sept 2018 | 17 Sept 2018 | 20 Sept 2018 | 27 Sept 2018 | 30 Sept 2018 | 4 Oct 2018 |
| | 5 Oct 2018 | 8 Oct 2018 | 10 Oct 2018 | 12 Oct 2018 | 13 Oct 2018 | 14 Oct 2018 |
| 2nd campaign | 26 Jun 2019 | 27 Jun 2019 | 28 Jun 2019 | 29 Jun 2019 | 30 Jun 2019 | 4 Jul 2019 |
| | 23 Jul 2019 | 24 Jul 2019 | 25 Jul 2019 | | | |

## 3 Measurement and validation data

### 3.1 Ground-based measurements

Model calibration and inversion is performed with PV power data recorded over two measurement campaigns in autumn 2018 and summer 2019, as part of the MetPVNet measurement campaign [see Chapter 3 of Meilinger et al. (2021)]. There were a total of 24 stations spread out in the region around Kempten (47.715924°N, 10.314006° E), as shown in Figure 3, with 22 of them equipped with silicon-based pyranometers measuring both GHI and GTI in the plane-of-array of the PV system. Two master stations (MS01 and MS02) were also equipped with secondary standard pyranometers and pyrheliometers in order to measure both components of the incoming short wave radiation, cloud cameras as well as spectrometers to record spectral information. The station MS01 also contained a sun photometer to determine aerosol properties, as part of AERONET, as well as a pyrgeometer to measure longwave downwelling irradiance.

The PV power data was for the most part provided by the local distribution network operator Allgäuer Überlandwerk GmbH (AÜW), recorded in 15 minute intervals. These data represent the amount of energy generated in the last 15 minutes, so that care needs to be taken to translate them into a measured power corresponding to a specific time stamp. For this purpose the data are simply shifted by half a period and resampled, since by integration of power over 15 minutes one effectively smooths the power curve. In addition there were five stations equipped by egrid GmbH with high frequency power measurement devices: for these stations the power was recorded in 1 Hz resolution.

Analysis of the measured data revealed a total of twelve clear sky days that occurred between 12 September 2018 and 14 October 2018, as well as nine clear sky days between 25 June 2019 and 13 August July 2019, as shown in Table 6. COSMO model data for the corresponding days was procured from Germany's National Meteorological Service, the Deutscher Wetterdienst (DWD), in order to accurately recreate atmospheric conditions using `cosmomystic`. These days are used for calibration of each PV system.

The network of PV systems was equipped with low cost silicon-based pyranometers, with two devices per station: one mounted in the plane of the PV array and one horizontal, with 1 Hz resolution and an overall accuracy of 5%. These sensors had been absolutely calibrated at the Leibniz Institute for Tropospheric Research (TROPOS) prior to the campaign, by comparing their output to that of a secondary standard pyranometer (2% accuracy). In order to compensate for errors in mounting the plane-of-array pyranometers, the calibration algorithm described in 2.2 is also applied to the pyranometer data, in this case without an optical model and only using data up to a SZA of 60 degrees. Due to the substantial cosine bias, a correction factor

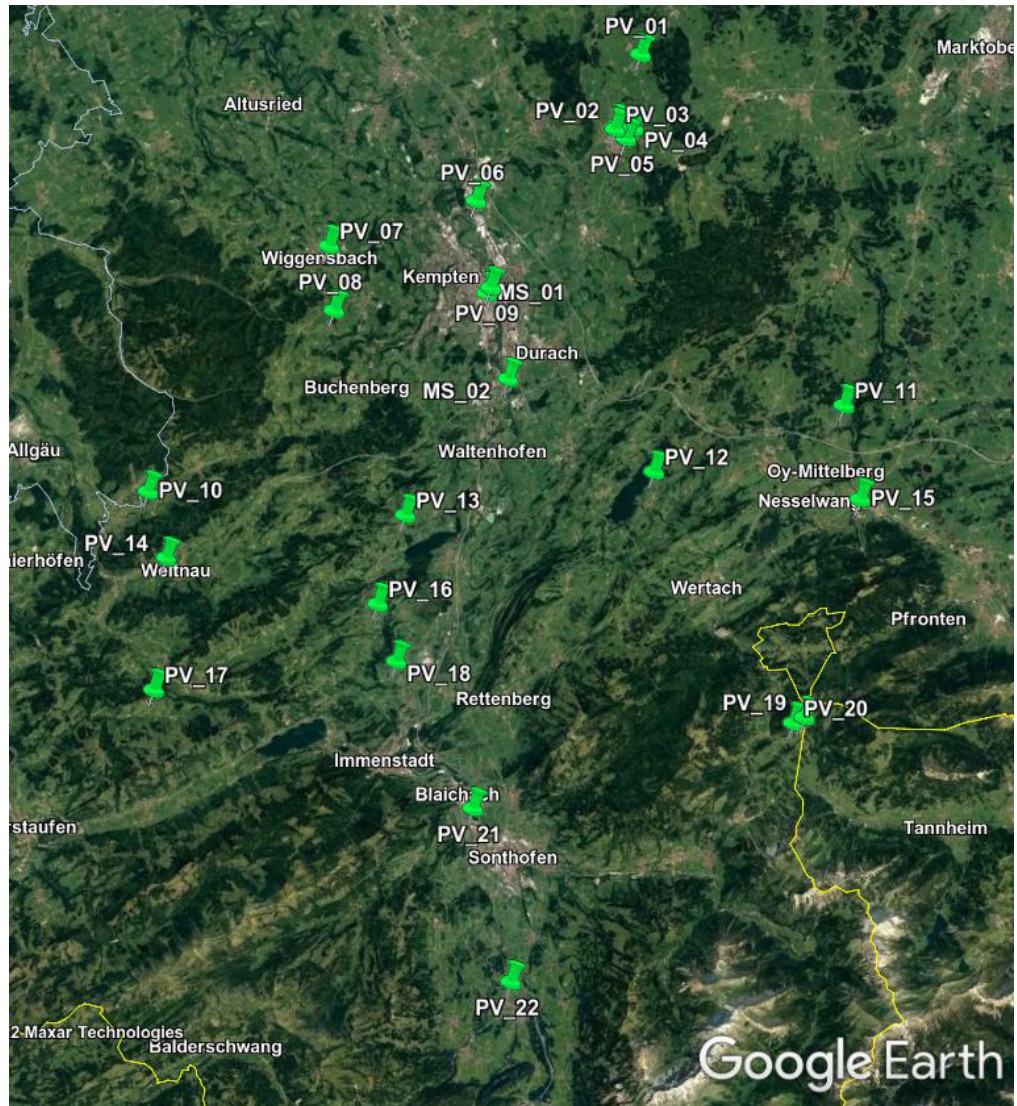

**Figure 3.** Map showing locations of PV systems used in the MetPVNet measurement campaign (taken from Google Earth). The top left corner is at 47.85° N, 10.09° E, the bottom right corner at 47.38° N, 10.52° E. The yellow line marks the border between Germany and Austria; the grey line is the border between the states of Bavaria and Baden-Württemberg.

is empirically determined:

$$C(\mu) = -3.01\mu^3 + 5.59\mu^2 - 3.34\mu + 1.45 \tag{14}$$

where $\mu = \cos\theta_0$ for horizontal sensors and $\mu = \cos\Theta$ for tilted sensors ($\theta_0$ is the SZA and $\Theta$ the angle of incidence, see Eq. A8). The pyranometer data is used for comparison with the inverted irradiance (both tilted and horizontal), as well as for finding atmospheric optical properties using the lookup table method.

In order to validate the PV- and pyranometer-based COD retrievals, it would be appropriate to use another ground-based source of cloud optical properties, however unfortunately there are no appropriate meteorological stations in the immediate area that could have been used for this purpose. Although there are several DWD stations in the Allgäu region (in Kempten, Oberstdorf and Hohenpeissenberg), these provide information on irradiance (direct and diffuse), but not on cloud optical properties [see Becker and Behrens (2012)]. Thus, a true validation would have to be done for another dataset with PV systems closer to a measurement station that has ground-based retrievals of COD. For this reason, the COD retrievals are simply compared to the corresponding cloud properties from weather models and satellite data.

## 3.2   Weather model data

The Consortium for Small-Scale Modelling (COSMO) numerical weather model is a nonhydrostatic regional model developed by the DWD (Baldauf et al., 2011). Note that this model was recently replaced by the so-called ICOsahedral Nonhydrostatic (ICON) model, which has been fully operational since January 2021. Since the measurement campaigns took place in 2018 and 2019, in this work the COSMO-EU model with a spatial resolution of 2.2km is used, both as input to the clear sky irradiance simulation (see Section 2.2.1), for PV model calibration (see Section 2.2.2) as well as for validation and/or comparison of the inverted irradiance with weather model predictions. For the clear sky simulation, temperature and pressure profiles as well as the water vapour column is extracted from COSMO data, whereas for both the calibration and inversion procedure the surface temperature and wind speed are used. For comparison and validation both direct and diffuse downward irradiance data is used.

For comparison with the cloud optical depths extracted from the PV systems, a two-dimensional COD field must be computed from the three-dimensional cloud variables generated by the COSMO model. For each grid cell, a cloud fraction variable in COSMO indicates which fraction of the cell is covered by clouds. To derive a vertically integrated COD, an assumption needs to be made as to how these clouds overlap in a model column. Following Scheck et al. (2018), the commonly used random-maximum cloud overlap assumption is adopted, along with the method of Matricardi (2005) in order to compute the vertically integrated COD for a number of subcolumns within each model column. From these subcolumn values a mean COD for the cloudy part of the column is derived. A total COD is then computed as the average of the column mean COD over $5 \times 5$ columns centred around the column containing the relevant ground station.

## 3.3   Satellite data

The Copernicus Atmospheric Monitoring Service (CAMS) radiation service (Qu et al., 2017; Schroedter-Homscheidt et al., 2022) is an online satellite and numerical model-based service with radiation, cloud and aerosol data available for free download, covering the period from February 2004 to the present. The spatial coverage is Europe/Africa/Middle East/Eastern part of South America/Atlantic Ocean, interpolated to the point of interest, and with a time resolution of up to one minute. In this work the global, direct and diffuse components of irradiance are imported from CAMS (version 4.0), for each station and for all days in the two measurement campaigns. These data are used as a comparison for the irradiance inverted from PV systems.

In addition to irradiance, CAMS provides data on cloud and aerosol properties. In this work, the cloud parameters from the AVHRR (Advanced Very High Resolution Radiometer) Processing scheme Over cLouds, Land and Ocean Next Generation

(APOLLO_NG) analysis (Kriebel et al., 2003; Klüser et al., 2015) are used, using data from the Spinning Enhanced Visible and InfraRed Imager (SEVIRI) instrument on board the Meteosat Second Generation (MSG) satellite. In this case the so-called Stephens method (Stephens et al., 1984) is used to determine the COD, using a two-stream solution of the radiative transfer equation, along with an updated algorithm using a probabilistic approach for cloud detection (Klüser et al., 2015). For comparison with the COD inferred from the PV systems, APOLLO_NG data is extracted for the closest pixel to each station.

## 4    Results

The calibration and inversion procedure described in Section 2 is applied to the data from the measurement campaign described in Section 3 in order to extract irradiance and optical properties from the PV systems in the Allgäu region. After a brief summary of the calibration results in Section 4.1, the retrievals of tilted and horizontal irradiance are presented in Sections 4.2 and 4.3, and the inferred COD results are shown in Section 4.4.

### 4.1    Model calibration and uncertainty

The PV models in Eqs. (1) and (2) are used together with the clear sky simulation described in Section 2.2.1 and the clear sky days (in Table 6) in order to calibrate each system. In each individual case those days that turned out not to be clear are discarded from the calibration dataset, and the data is restricted to those time periods in which the required inputs such as ambient temperature, atmospheric longwave irradiance and wind speed are available. In order to validate the calibration results the retrieved elevation and azimuth angles are compared to ground truth data from the Bavarian Agency for Digitisation, High-Speed Internet and Surveying (LDBV). The so-called "Level of Detail 2" (LoD2) database contains a 3D building model constructed using airborne laser scanning, so that the roof pitch of individual buildings can be extracted. Figure 4 shows a comparison of the retrieved orientation angles with the ground truth values, for each system and using the linear temperature model.

In most cases the algorithm finds reasonable values for the angles: the larger deviations can usually be explained individual cases, for instances for PV04 the inverter MPP tracking algorithm distorts the clear sky days, whereas for the systems at PV11 the different PV arrays at the site are not well characterised. In other cases shading effects played a role: in most cases the calibration performed better when using both summer and autumn data, since in summer the sun is much higher and shading effects play a smaller role. In general the model calibration works best when using as much data as possible, since one has for instance more variation in temperature in order to find more reliable temperature model parameters.

As discussed in Section 2.2.2, several parameters are correlated with each other: the size of the PV system (captured by the factor $s$) correlates with the tilt angle $\theta$, whereas the azimuth angle $\phi$ shows a large correlation with the parameters of the temperature model, since the warming up and cooling down of the PV system is delayed with respect to the diurnal variation of solar irradiance. In general the use of measured module temperature leads to better calibration results. It turns out that the calibration algorithm presented here doesn't perform well using the non-linear Faiman temperature model and 15-minute power data, even though this model couples irradiance and windspeed in a more physically correct way [see for instance

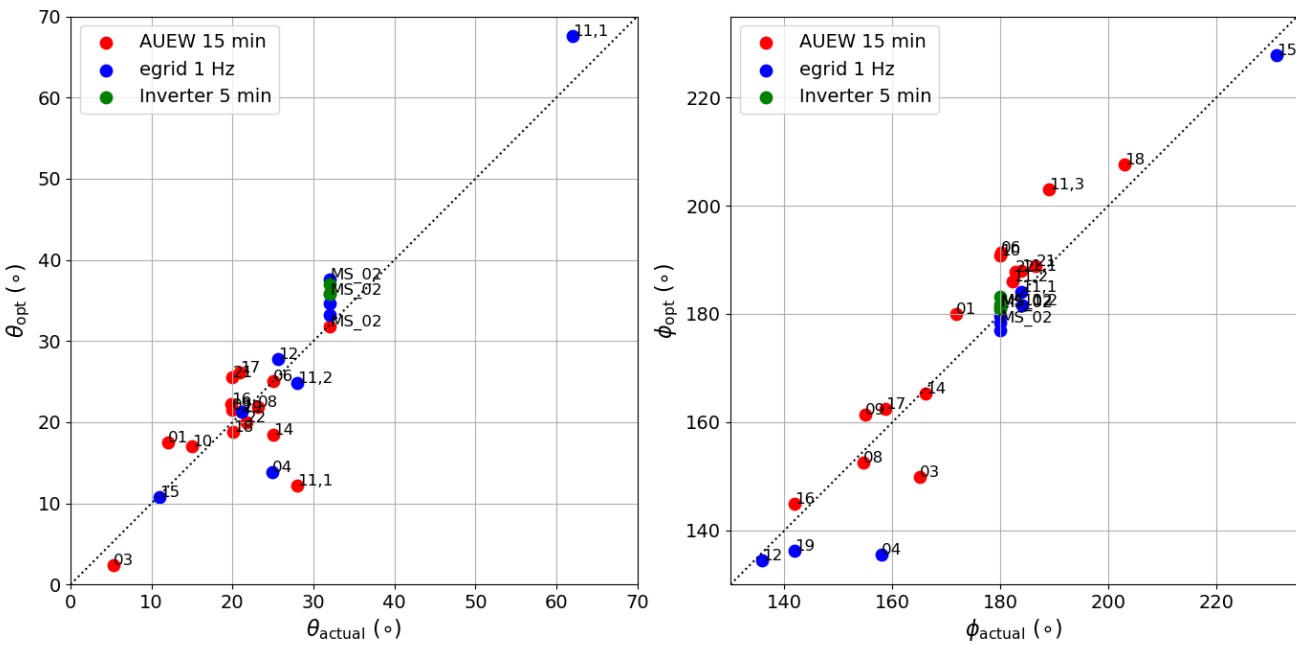

**Figure 4.** Comparison of retrieved azimuth (elevation) angle $\phi_{\mathrm{opt}}$ ($\theta_{\mathrm{opt}}$) with ground truth ($\phi_{\mathrm{actual}}$ and $\theta_{\mathrm{actual}}$) from airborne laserscanning data.

Faiman (2008); Barry et al. (2020)]. The benefit of this model is lost for coarsely resolved 15-minute data, so that in the end
the algorithm proposed here does not always find an optimal solution, specifically if the temperature model parameters are unknown. The bias that then occurs in the final tilted irradiance inversion results can be seen in the plots in Section 4.2 as well as in the results in Tables 8 and 9. However, this bias in the tilted irradiance does not always translate into a bias in GHI, as will be seen below. Table 7 lists the PV systems used in this work, along with the corresponding time resolution of their data.

### 4.2   Global tilted irradiance from PV power data

In this section the plane-of-array irradiance from PV power retrievals is compared to the tilted pyranometer measurements at selected stations during the two measurement campaigns. The results are obtained using two different approaches for module temperature: (i) the linear temperature model [Eq. (A3)], and (ii) the non-linear Faiman temperature model [Eq. (A4)]. The results are compared in Tables 8 and 9, and scatter plots for both models are shown. Both stations with 15-minute power data as well as those with high frequency data (1 Hz data, smoothed to 1-minute resolution) are included in the analysis, and in
each case a comparison is made to the measurements from TROPOS silicon-based pyranometers, except for the master station MS02, where the tilted Kipp&Zonen CMP11 pyranometer is used for validation. In all cases a limit of 80 degrees is imposed on both the solar zenith angle and the incident angle, in order to avoid possible errors from both the radiative transfer simulation as well as the optical model.

**Table 7.** List of PV systems used for this work (see also the map in Fig. 3). The data resolution column indicates whether a particular system is used for this analysis or not, with an explanation given in case the system is omitted. Note that stations MS01 and PV02 had no PV systems, only pyranometers and other measurement equipment. PV11 has four separate PV systems.

| Station | Data resolution | | Mounting | Comments |
|---|---|---|---|---|
| | **2018** | **2019** | | |
| MS02 | 15 min | 15 min / 1 s | Ground | |
| PV01 | 15 min | 15 min | Rooftop | |
| PV03 | - | - | Rooftop | Calibration errors |
| PV04 | - | - | Rooftop | Calibration errors |
| PV05 | - | - | Rooftop | No data |
| PV06 | 15 min | 15 min | Ground | |
| PV07 | - | - | Rooftop | Calibration errors |
| PV08 | 15 min | - | Rooftop | No data in 2019 |
| PV09 | - | - | Rooftop open | Calibration errors |
| PV10 | 15 min | 15 min | Ground | |
| PV11,1 | 15 min | 1 s | Rooftop | |
| PV11,2 | 15 min | 15 min | Rooftop | |
| PV11,3 | 15 min | 15 min | Rooftop | |
| PV11,4 | 1 s | - | Ground | No data in 2019 |
| PV12 | 1 s | 1 s | Rooftop | |
| PV13 | - | - | Rooftop | No 2018 data, calibration problems |
| PV14 | 15 min | 15 min | Rooftop open | |
| PV15 | 1 s | 1 s | Rooftop | |
| PV16 | 15 min | 15 min | Rooftop | |
| PV17 | 15 min | 15 min | Rooftop | |
| PV18 | 15 min | 15 min | Rooftop | |
| PV19 | 1 s | 1 s | Rooftop | |
| PV20 | - | - | Rooftop | No data |
| PV21 | 15 min | 15 min | Rooftop | |
| PV22 | - | - | Rooftop | Calibration errors |

Figures 5 and 6 show a comparison between the retrieved GTI and that measured by pyranometers, for the 1-minute and 15-minute data respectively, for each measurement campaign and using both the linear and non-linear temperature models. The corresponding statistical measures of mean bias error (MBE), defined by

$$\text{MBE} = \frac{1}{n} \sum_{i=1}^{n} (X_{\text{inv}} - X_{\text{ref}}), \tag{15}$$

**Table 8.** Mean bias error (in W m$^{-2}$) and relative mean bias error (in brackets in %) of GTI retrievals, compared to tilted pyranometers. The values marked with * are high due to calibration errors using the non-linear temperature model with 15-minute data.

|  | **2018** | | **2019** | |
| --- | --- | --- | --- | --- |
|  | Linear | Non-linear | Linear | Non-linear |
| **1min** | 16.23 (3.2) | 5.29 (1.0) | 21.12 (4.1) | 12.25 (2.4) |
| **15min** | 28.74 (5.9) | 87.55* (18.0)* | 40.20 (7.8) | 96.40* (18.6)* |

as well as root mean squared error (RMSE), defined by

$$\mathrm{RMSE} = \sqrt{\frac{1}{n}\sum_{i=1}^{n}(X_{\mathrm{inv}} - X_{\mathrm{ref}})^2}, \tag{16}$$

for the inverted quantities $X_{\mathrm{inv}}$ and the reference quantities $X_{\mathrm{ref}}$ are shown in Tables 8 and 9, along with the relative error metrics rMBE and rRMSE calculated by normalising the MBE and RMSE with the mean of the reference quantity, $\langle X_{\mathrm{ref}} \rangle$. The scatter plots throughout this work are coloured according to a probability density function calculated using the multi-variate Gaussian kernel density function "gaussian_kde" in the Python toolbox scipy, with yellow (light grey) for high and blue (dark grey) for low frequency points in the colour (black and white) version. In Figs. 5 and 6 one can see that most points lie close
to the 1:1 line, for both campaigns, albeit with a positive bias in all cases. The 1-minute data shows a slightly larger spread of points than the 15-minute data, since in the former case there are more outliers caused by i) temperature effects, ii) 3D radiative transfer effects, and iii) spatial effects due to differences in cloud cover and sensor position between PV and pyranometer. In addition the slightly different geometry of flat PV arrays compared to glass dome-shaped pyranometers could play a role, especially when it comes to their sensitivity to different viewing angles. Another possible reason for the positive bias could be
a systematic bias in the tilted pyranometer measurements, even after the bias correction described in Section 3.1.

The two different temperature models achieve similar results for the 1-minute data, with the non-linear model showing an MBE of 5.29 W m$^{-2}$ (12.25 W m$^{-2}$) in autumn 2018 (summer 2019), compared to an rMBE of 16.23 W m$^{-2}$ (21.12 W m$^{-2}$) for the linear model. In general the algorithm performs worse with 15-minute data, which has to do with errors from the calibration procedure as well as uncertainties in the PV power measurements – the systems with high frequency measurements
are thus in general better characterised and deliver more accurate irradiance retrievals, as shown in Section 4.3 below. This effect is quite extreme for the non-linear Faiman temperature model (see the values marked with * in Tables 8 and 9 as well as the plots in the lower panels of Fig. 6), since in some cases the calibration algorithm cannot find an optimal solution and the a-priori values have to be relied upon, leading to an average MBE of 91.98 W m$^{-2}$.

### 4.3 Global horizontal irradiance from PV power data

In the following the global tilted irradiance (GTI) retrievals are converted to global horizontal irradiance (GHI) and compared to the measurements from pyranometers as well as to the satellite and weather model data. This conversion is performed in

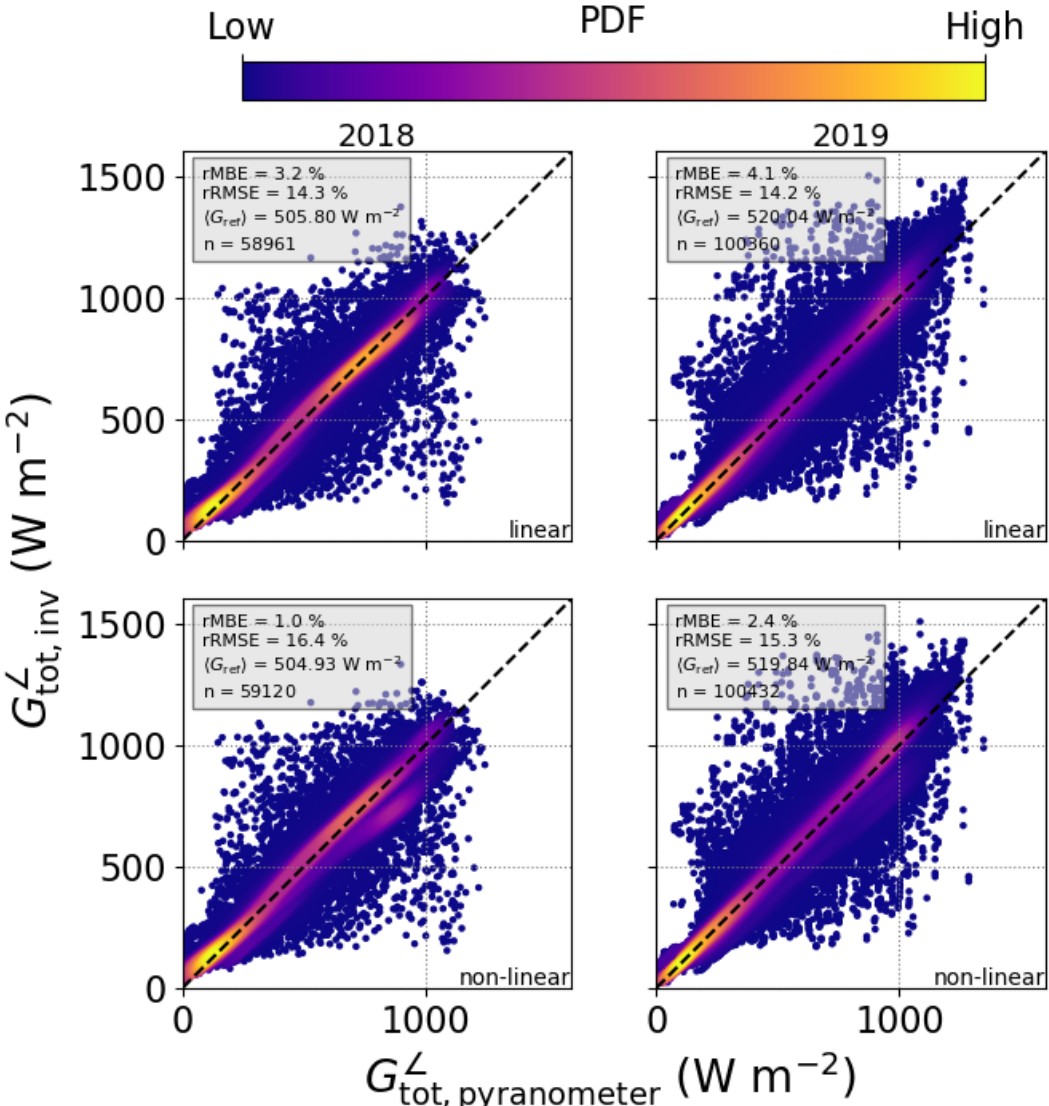

**Figure 5.** Combined comparison of GTI retrieved from PV power measurements with that measured by tilted pyranometers, using data in 1-minute resolution from MS02, PV11, PV12, PV15, PV19 (cf. Table 7), for 2018 (left) and 2019 (right), together with the linear (top) and non-linear (bottom) temperature models. Relative mean bias error (rMBE) and relative root mean squared error (rRMSE) are shown in the inset, along with the mean of the reference GTI as well as the number of data points used, denoted as $\langle G_{\mathrm{ref}} \rangle$ and $n$, respectively.

three different ways, depending on the prevailing weather conditions, as described in Section 2.3. In case of clear skies, the DISORT-based LUT is used to find the value of aerosol optical depth, and the inferred AOD is then used to calculate the direct and diffuse horizontal irradiance components, which result in the GHI. The same method is used under cloudy skies using the
DISORT-based COD LUT, but only for times at which the sensor is under a cloud, using the inferred cloud fraction method as

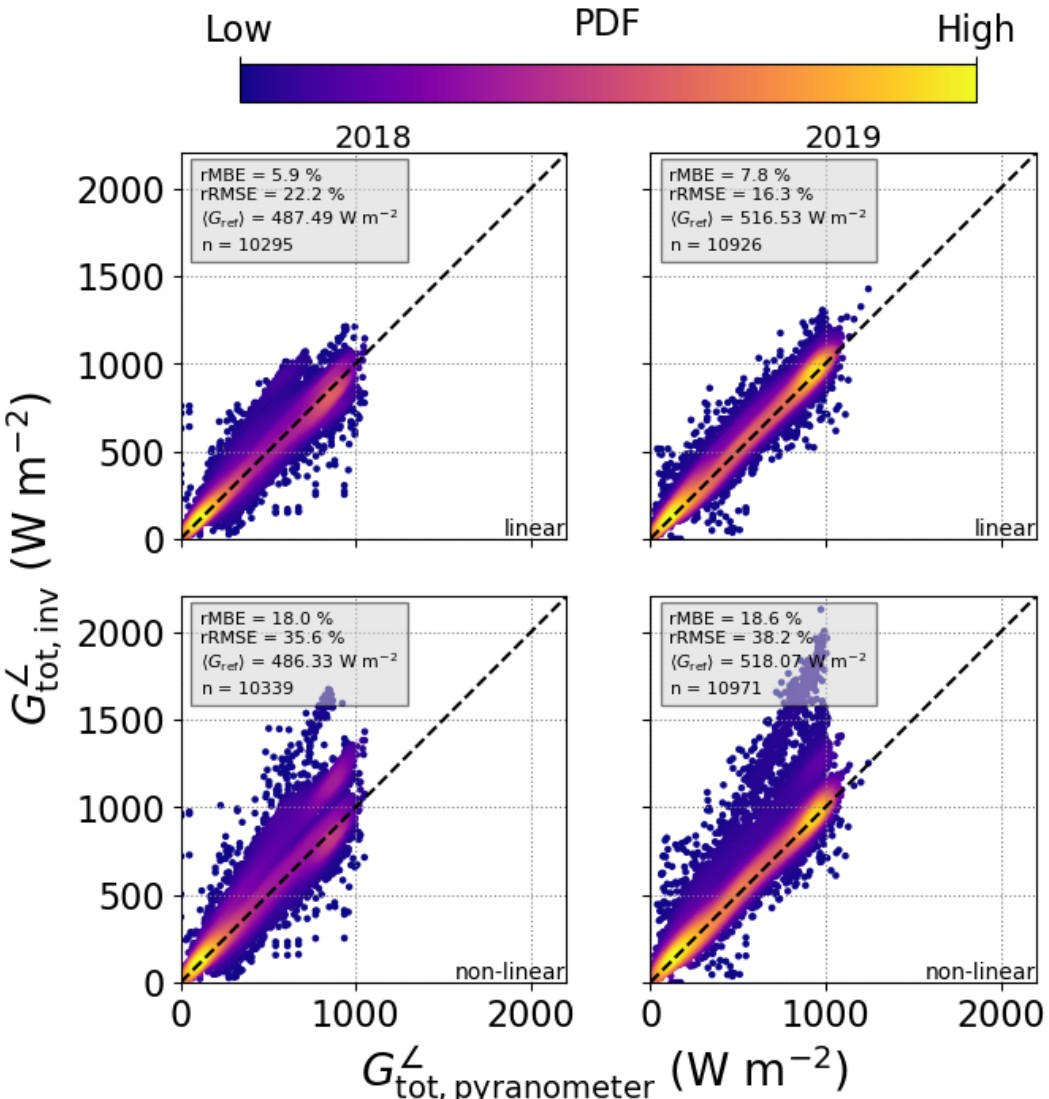

**Figure 6.** Same as Fig. 5 using data in 15-minute resolution from MS02, PV01, PV06, PV08, PV10, PV11, PV14, PV16, PV17, PV18, PV21 (cf. Table 7) for 2018 (left) and 2019 (right). Note that the large errors in the non-linear model in the bottom row result from errors in the calibration procedure (see the values marked with * in Tables 8 and 9).

described in Section 2.3.3. This means that it may tend to underestimate the GHI under broken cloud conditions, which will be discussed in detail below.

The third approach for finding the GHI is using the MYSTIC-based LUT described in Section 2.3.5, where the input parameters to the LUT are simply array geometry, sun position and cloud fraction. In this case there are certain restrictions on these parameters, as shown in detail in Table 5. This means that not all of the retrieved GTI data points can be transformed into GHI

**Table 9.** Root mean squared error (in W m$^{-2}$) and relative RMSE (in brackets in %) of GTI retrievals, compared to tilted pyranometers. The values marked with * are high due to calibration errors using the non-linear temperature model with 15-minute data.

| | 2018 | | 2019 | |
|---|---|---|---|---|
| | Linear | Non-linear | Linear | Non-linear |
| **1min** | 72.34 (14.3) | 82.68 (16.4) | 73.97 (14.2) | 79.71 (15.3) |
| **15min** | 108.27 (22.2) | 172.96* (38.1)* | 83.94 (16.3) | 197.83* (38.2)* |

**Table 10.** Mean bias error and root mean squared error (in W m$^{-2}$), along with rMBE and rRMSE (in brackets in %) of GHI retrievals using the 1D DISORT (AOD and COD) and the 3D MYSTIC LUTs, compared to horizontal pyranometers, for 1-minute and 15-minute data.

| Data | Measure | LUT | 2018 | | 2019 | |
|---|---|---|---|---|---|---|
| | | | Linear | Non-linear | Linear | Non-linear |
| **1min** | **MBE** | 1D AOD | 18.15 (3.9) | 1.83 (0.4) | 9.44 (1.4) | 9.75 (1.5) |
| | | 1D COD | 18.32 (8.2) | 16.68 (7.4) | 13.85 (4.7) | 8.73 (2.9) |
| | | 3D GHI | 29.90 (5.6) | 0.60 (0.1) | 20.69 (3.1) | 3.39 (0.5) |
| | **RMSE** | 1D AOD | 30.17 (6.5) | 33.55 (7.3) | 33.81 (5.1) | 35.24 (5.3) |
| | | 1D COD | 63.10 (28.2) | 64.70 (28.6) | 66.72 (22.7) | 70.38 (23.3) |
| | | 3D GHI | 89.76 (16.8) | 87.89 (16.3) | 102.69 (15.4) | 111.28 (16.5) |
| **15min** | **MBE** | 1D AOD | 10.01 (2.3) | 1.76 (0.4) | 1.75 (0.3) | 6.44 (1.0) |
| | | 1D COD | 37.34 (15.5) | 40.65 (17.5) | 46.39 (15.6) | 44.55 (15.4) |
| | **RMSE** | 1D AOD | 38.96 (9.0) | 39.59 (9.5) | 35.93 (5.2) | 36.89 (5.5) |
| | | 1D COD | 78.32 (32.5) | 82.74 (35.6) | 91.96 (30.8) | 100.71 (34.8) |

using this method – in particular the SZA is limited to between 20° and 60°, and the cloud fraction to between 0.13 and 0.82, so that neither completely overcast nor clear sky conditions are taken into account. This method thus deals with the case of mixed/broken cloud conditions, in which there are more likely to be errors due to 3D effects and sensor position.

### 4.3.1 GHI retrieval validation with pyranometer measurements

As discussed in Section 4.2 above, the PV systems with 1-minute data show the best calibration results and the most accurate tilted irradiance retrievals. The scatter plots in Fig. 7 compare the GHI retrieved from these systems to that measured by horizontal pyranometers, using all three methods and for both temperature models. The statistical measures of the different retrievals are shown in Table 10, where it can be seen that the mean bias error reaches the goal of 5 W m$^{-2}$ only in certain cases.

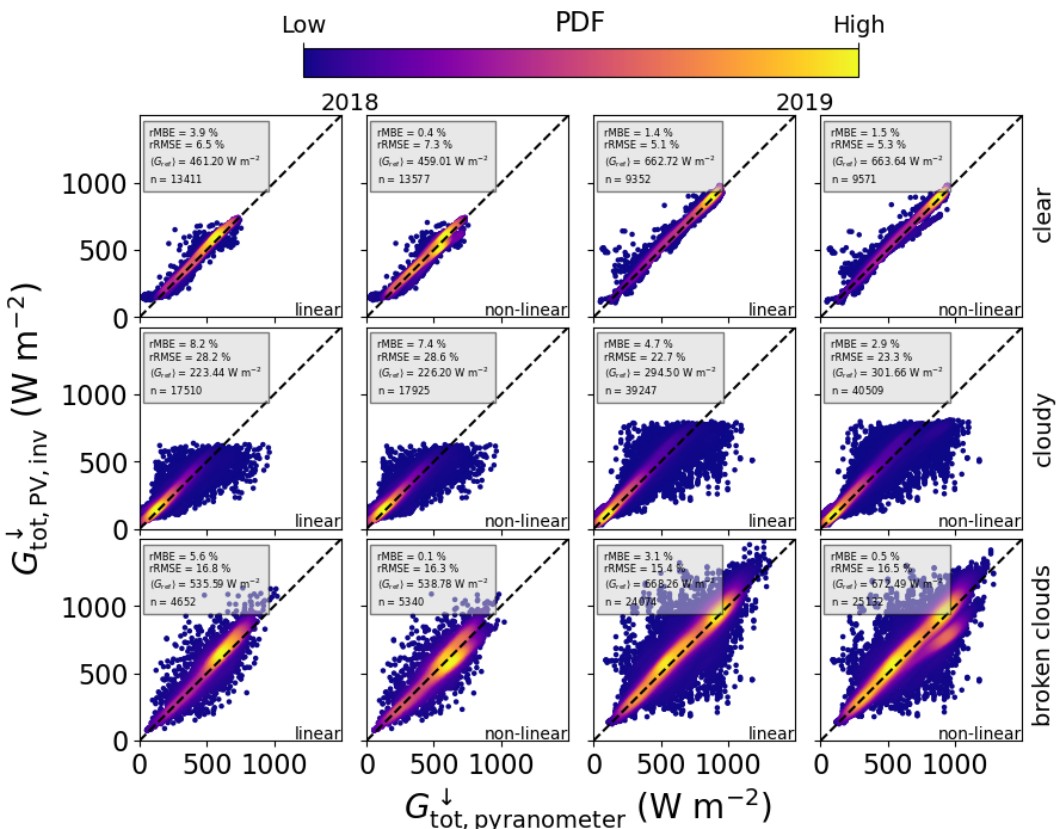

**Figure 7.** Combined comparison of GHI retrieved from PV power measurements with that measured by horizontal pyranometers, using data in 1 minute resolution from MS02, PV11, PV12, PV15, PV19, for 2018 (left two columns) and 2019 (right two columns), and both the linear and non-linear temperature models. The top row shows GHI retrieved via AOD under clear skies using the DISORT AOD LUT, the middle row is for cloudy periods via the COD using the DISORT COD LUT, and the bottom row is for broken cloud periods using the MYSTIC 3D LUT.

Under clear conditions (top row of Fig. 7), the linear model applied to 1-minute PV data achieves an rMBE for the GHI of 18.15 W m$^{-2}$ (3.9%) in autumn 2018 and 9.44 W m$^{-2}$ (1.4%) in summer 2019, respectively. Note that the mean irradiance is higher in summer, but there are less points that can be classified as clear ($n \simeq 9400$ compared to $n \simeq 13400$ in autumn). The non-linear model performs significantly better in autumn, with an rMBE for the GHI of 1.83 W m$^{-2}$ (0.4%), but in summer the bias is similar to the linear model [9.75 W m$^{-2}$ (1.5%)]. Interestingly the linear temperature model performs better in summer

than in autumn, whereas the non-linear model performs better in autumn. This could be attributed to differences or uncertainties in the calibration of the temperature models. In general these results show that the DISORT LUT method performs comparably well for extracting GHI from PV power measurements under clear conditions. In all cases the rRMSE is of the order of 5% to 7%, on average 33.19 W m$^{-2}$.

It is evident from the middle row of Fig. 7 that the bias is greater under cloudy skies, with an average over both campaigns of 11.29 W m$^{-2}$ for the non-linear model and 17.50 W m$^{-2}$ for the linear model. In autumn the lower average irradiance of 225 W m$^{-2}$ leads to a higher relative MBE than in summer, where the average irradiance is 298 W m$^{-2}$. At this point it is worth mentioning that the algorithm only finds the COD and thus the GHI when the sensor is under a cloud, hence the lower average irradiance in comparison with that under clear skies, and the RMSE is higher (66.23 W m$^{-2}$ on average) than in the case of clear skies, where it is 33.19 W m$^{-2}$ on average, as expected. This also means that averaging these results over 60 minutes can lead to erroneous values for the irradiance, especially under broken clouds, since the periods of cloud enhancement within each one hour window will not be taken into account.

The GHI retrieved from the 3D MYSTIC LUT shows significantly lower bias in the case of the non-linear temperature model (2.00 W m$^{-2}$ compared to 12.71 W m$^{-2}$ using the DISORT COD method, averaged over both campaigns), but in the case of the linear temperature model the bias is higher (25.30 W m$^{-2}$ compared to 16.09 W m$^{-2}$). The good performance of the non-linear model could indeed be a result of the improved treatment of the PV module temperature during broken cloud conditions: although neither model contains a dynamic term, the non-linear model couples irradiance and windspeed in a more physically correct way. However, it must be noted that due to the restrictions on the MYSTIC LUT (see Table 5), the number of inferred irradiances included in the statistics are far lower than for the COD method. Another confounding factor could be that the PV panels show better efficiency at higher irradiance, and since the MYSTIC method also takes overshoots into account the irradiance is on average higher: 603.78 W m$^{-2}$ compared to 261.45 W m$^{-2}$ for the COD method. The larger fluctuations of irradiance under broken cloud conditions also lead to a larger RMSE than for the case of cloudy skies, and in summer the RMSE is the highest, as expected.

Figure 8 shows the GHI retrievals from the AÜW systems with 15-minute PV power measurements, under clear (top row) and cloudy (bottom row) skies. In this case the MYSTIC 3D LUT is not used, since the determination of cloud fraction with coarsely resolved data leads to erroneous results, and the rapid fluctuations of irradiance under broken clouds is not properly captured at 15-minute resolution. The DISORT 1D LUT performs well under clear skies, as to be expected, with the linear model showing an average MBE of 5.88 W m$^{-2}$ and the non-linear model 4.10 W m$^{-2}$. Once again the non-linear model outperforms the linear one in autumn 2018, but this trend is reversed in summer 2019. This systematic effect is most probably due to uncertainties in the temperature model calibration. Under cloudy skies the GHI retrievals show a significant positive bias of on average 41.87 W m$^{-2}$ (42.60 W m$^{-2}$) for the linear (non-linear) model, which means that the retrieved COD is too small. This will be discussed further in Section 4.4. Interestingly the large bias errors in tilted irradiance for the non-linear model are not evident in the horizontal irradiance results, which is probably due to the fact that far less points are taken into account in the statistics for GHI (compare the values of $n$ in Figs. 8 and 6). In other words, the outliers have been removed by selecting either clear sky days or periods for which the cloud fraction is 100%.

### 4.3.2 Comparison to satellite and weather model irradiance data

One of the main aims of this work is to show that PV systems can provide a reliable source of information on global horizontal irradiance that is complementary to that from satellite and weather models. Figs 9 and 10 (with corresponding statistical

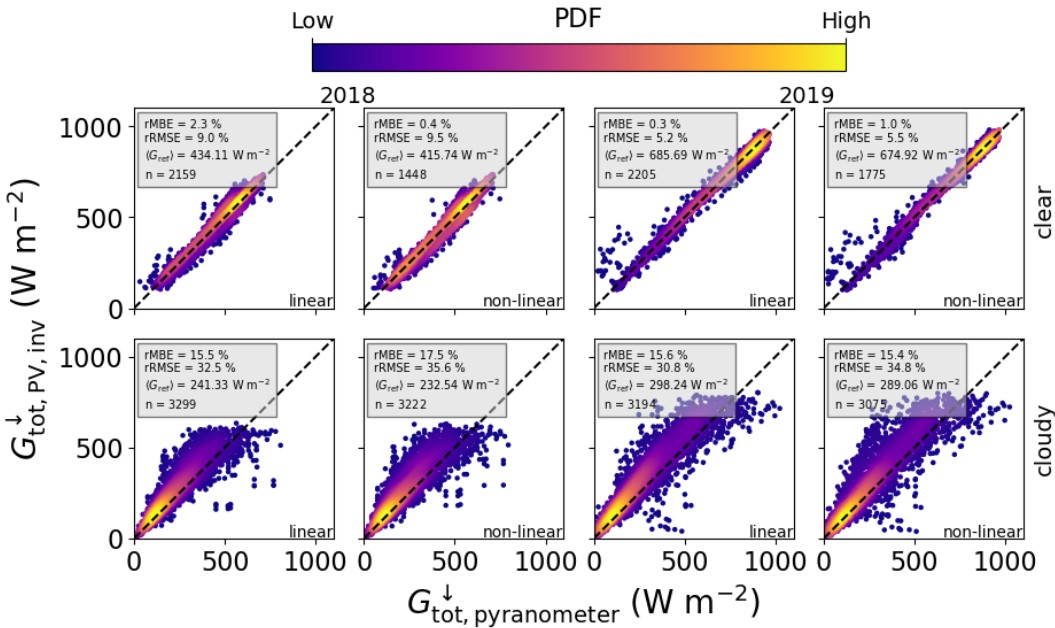

**Figure 8.** Combined comparison of GHI retrieved from PV power measurements with that measured by horizontal pyranometers, using data in 15-minute resolution from MS02, PV01, PV06, PV08, PV10, PV11, PV14, PV16, PV17, PV18, PV21, for 2018 (left two columns) and 2019 (right two columns), and both the linear and non-linear temperature models. The top row shows GHI retrieved via AOD under clear skies using the DISORT AOD LUT, the bottom row is for cloudy periods via the COD using the DISORT COD LUT.

measures in Table 11) show the comparison between GHI retrieved from PV power using the aerosol and cloud optical depths and that from the CAMS retrieval, for 1-minute and 15-minute power data, respectively. Under clear sky conditions the retrieved

GHI shows an average MBE of 15.95 W m$^{-2}$ for the linear model and 7.54 W m$^{-2}$ for the non-linear model. These values are similar to those found by comparing with ground-based pyranometers, confirming the accuracy of the CAMS data. On the other hand, the GHI retrieved using the DISORT COD LUT under cloudy skies shows a significant negative bias compared to the CAMS retrieval (-37.77 W m$^{-2}$ in autumn and -86.87 W m$^{-2}$ in summer). There are two possible reasons for this: firstly the simplification to one cloud type means that thinner clouds or multi-layer cloud situations are not properly represented in

the model (see the discussion on COD retrievals in Section 4.4 below). However, the main reason is related to the retrieval algorithm: by only considering measurements where the PV system is under a cloud, only those periods with lower irradiance values are retrieved and overshoots are ignored, so that at 1-minute resolution a large negative bias in irradiance is found. Since the CAMS irradiance retrieval is based on the Heliosat-4 method, in which cloud properties from the APOLLO_NG method are updated every 15 minutes (Qu et al., 2017), one should expect this bias to reduce at coarser resolution. Indeed, the 15-minute

data in the bottom row of Fig. 10 confirms this: the average bias is reduced to -3.73 W m$^{-2}$ in autumn and -15.07 W m$^{-2}$ in summer. Note that an averaging to 60 minutes is not performed, due to the limitations of the DISORT COD algorithm, as discussed in Section 4.3.1.

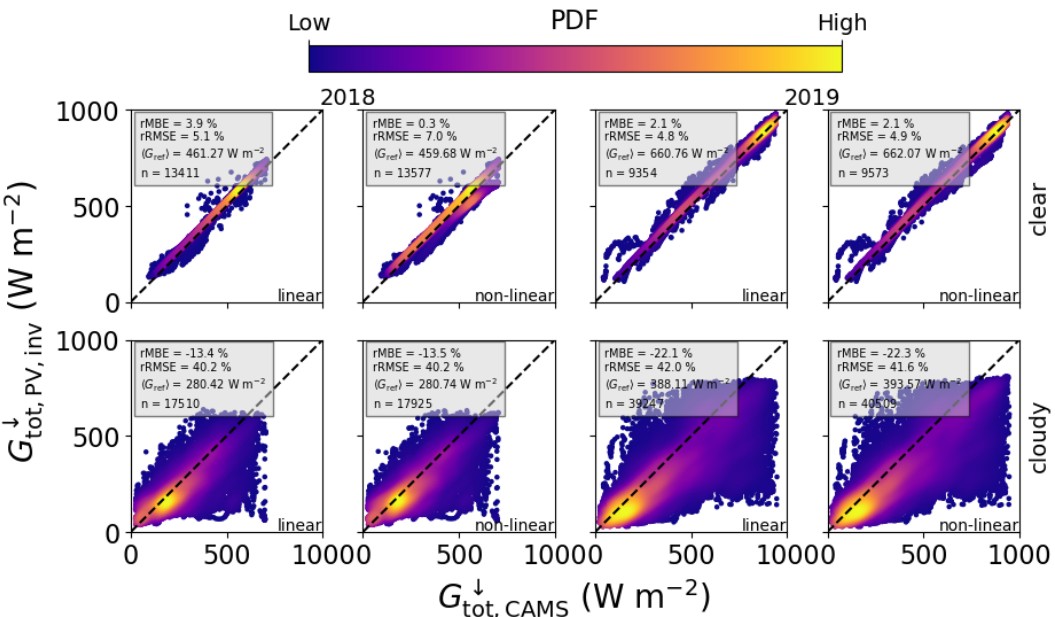

**Figure 9.** Combined comparison of GHI retrieved from PV power measurements with that from CAMS, using data in 1-minute resolution from MS02, PV11, PV12, PV15, PV19, for 2018 (left two columns) and 2019 (right two columns), and both the linear and non-linear temperature models. The top row shows GHI retrieved via AOD under clear skies using the DISORT AOD LUT, the bottom row is for cloudy periods via the COD using the DISORT COD LUT.

**Table 11.** Mean bias error and root mean squared error (in W m$^{-2}$), along with rMBE and rRMSE (in brackets in %) of GHI retrievals using 1D DISORT (AOD and COD) and the 3D MYSTIC LUT, compared to CAMS, for 1-minute and 15-minute data.

| Data | Measure | LUT | 2018 | | 2019 | |
|------|---------|-----|------|------|------|------|
| | | | Linear | Non-linear | Linear | Non-linear |
| 1min | MBE | 1D AOD | 18.07 (3.9) | 1.16 (0.3) | 13.83 (2.1) | 13.91 (2.1) |
| | | 1D COD | -37.68 (-13.4) | -37.86 (-13.5) | -85.85 (-22.1) | -87.88 (-22.3) |
| | RMSE | 1D AOD | 23.51 (5.1) | 32.19 (7.0) | 31.99 (4.8) | 32.48 (4.9) |
| | | 1D COD | 112.68 (40.2) | 112.78 (40.2) | 163.15 (42.0) | 163.61 (41.6) |
| 15min | MBE | 1D AOD | 5.74 (1.3) | -1.25 (-0.3) | 1.31 (0.2) | 5.78 (0.9) |
| | | 1D COD | -6.09 (-2.1) | -1.37 (-0.5) | -13.70 (-3.8) | -16.44 (-4.7) |
| | RMSE | 1D AOD | 28.74 (6.5) | 30.95 (7.3) | 29.26 (4.3) | 25.43 (3.8) |
| | | 1D COD | 75.78 (26.6) | 79.69 (29.0) | 114.36 (32.1) | 117.01 (33.6) |

The comparison with COSMO model data is shown in Fig. 11 and Table 12, where in this case the data is averaged over a 60 minute period. It is evident that the COSMO model shows a bias under clear sky conditions: here the assumed AOD is too

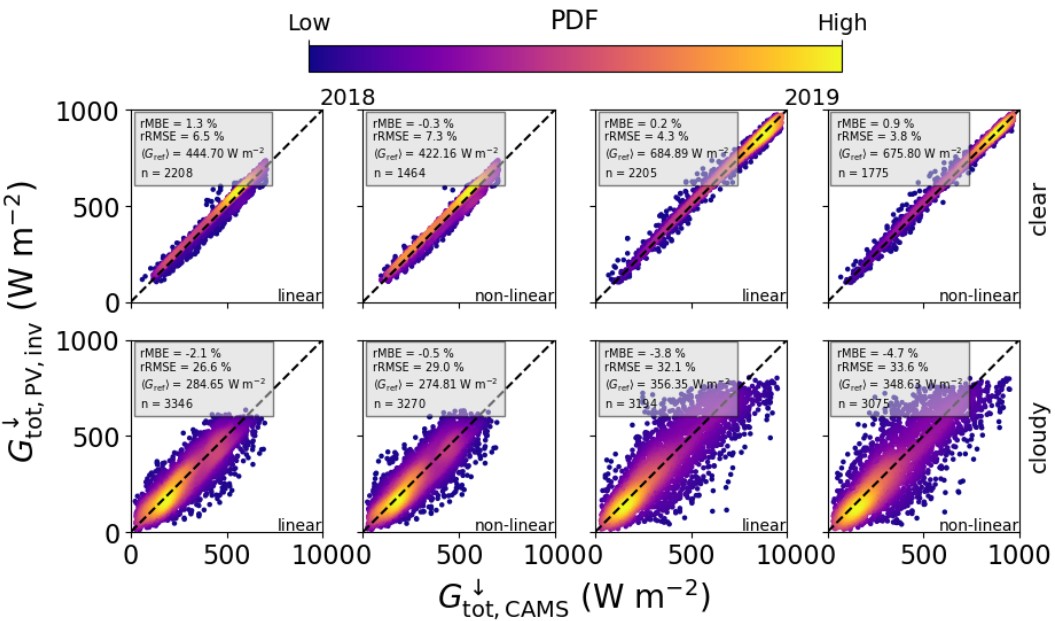

**Figure 10.** Combined comparison of GHI retrieved from PV power measurements with from CAMS, using data in 15-minute resolution from MS02, PV01, PV06, PV08, PV10, PV11, PV14, PV16, PV17, PV18, PV21, for 2018 (left two columns) and 2019 (right two columns), and both the linear and non-linear temperature models. The top row shows GHI retrieved via AOD under clear skies using the DISORT AOD LUT, the bottom row is for cloudy periods via the COD using the DISORT COD LUT.

**Table 12.** Mean bias error and root mean squared error (in W m$^{-2}$), along with rMBE and rRMSE (in brackets in %) of 60 minute average GHI retrievals using 1D DISORT (AOD and COD) and the 3D MYSTIC LUT compared to COSMO model data.

| Data | Measure | LUT | 2018 | | 2019 | |
|---|---|---|---|---|---|---|
| | | | Linear | Non-linear | Linear | Non-linear |
| **60 min average** | **MBE** | 1D AOD | 65.90 (16.2) | 57.72 (14.7) | 58.74 (9.9) | 61.33 (10.5) |
| | | 1D COD | 14.34 (5.2) | 16.77 (6.1) | -37.50 (-9.4) | -39.21 (-9.8) |
| | **RMSE** | 1D AOD | 72.71 (17.9) | 67.70 (17.2) | 65.65 (11.1) | 67.32 (11.5) |
| | | 1D COD | 124.16 (44.8) | 125.12 (45.4) | 143.86 (36.1) | 144.96 (36.4) |

high so that the irradiance turns out to be too small, with an average bias of 60.92 W m$^{-2}$. On the other hand, under cloudy conditions and especially under low light conditions in summer the irradiance from COSMO is larger than that retrieved from PV plants, which means that the COD in COSMO is too small. Here the average bias is -38.36 W m$^{-2}$. These results confirm the findings of Frank et al. (2018); Zubler et al. (2011), and are discussed further in connection with the cloud optical depth in Section 4.4 below.

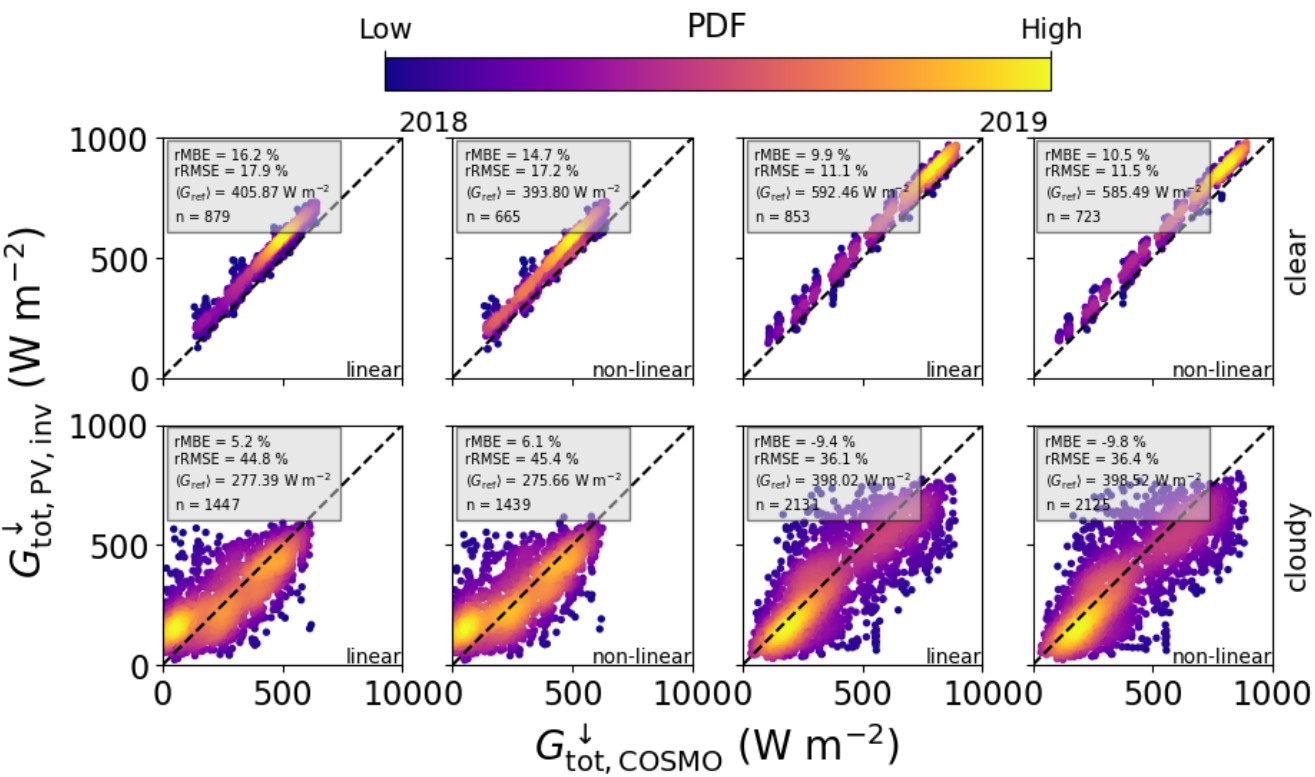

**Figure 11.** Combined comparison of GHI retrieved from 60 minute averaged PV power measurements under clear (top row) or completely cloudy (bottom row) conditions using the optical depth via DISORT LUT with that from the COSMO model, for all stations and for 2018 (left two columns) and 2019 (right two columns), using both the linear and non-linear temperature models.

## 4.4 Cloud optical depth retrievals

As discussed in Section 2.3.3, the cloud optical depth is retrieved from both PV systems and pyranometers, using a DISORT-based LUT. In order to avoid errors due to 3D effects, in this work only data with a cloud fraction of 1 are considered, in other words only completely overcast conditions. The results for the linear temperature model are shown in Figs. 12 and 13, compared to the APOLLO_NG and COSMO data respectively. As can also be seen in Tables 13 and 14, in most cases a smaller COD is extracted from PV systems, except for the comparison between the summer campaign and COSMO data, in which case a positive mean bias of approximately COD = 8 is found. Overall, the COD is mostly in the range between 1 and 10, for both campaigns. Taken at face value, the negative bias with respect to APOLLO_NG would imply a positive bias in GHI with respect to CAMS, which is not seen in the 1- and 15-minute retrievals. However these results cannot be directly compared, due to the effect of both spatial and temporal averaging as well as the limitation of the DISORT COD LUT algorithm, which ignores 3D effects.

**Table 13.** Mean bias error and root mean squared error (in W m$^{-2}$), along with rMBE and rRMSE (in brackets in %) of COD retrievals from PV systems, compared to the APOLLO_NG data.

| Data | Measure | 2018 | | 2019 | |
|------|---------|------|------|------|------|
| | | Linear | Non-linear | Linear | Non-linear |
| **60min average** | **MBE** | -3.22 (-25.0) | -3.57 (-27.2) | -3.58 (-15.4) | -3.24 (-13.6) |
| | **RMSE** | 15.20 (117.9) | 15.66 (119.3) | 18.99 (81.7) | 19.30 (81.2) |

**Table 14.** Mean bias error and root mean squared error (in W m$^{-2}$), along with rMBE and rRMSE (in brackets in %) of COD retrievals from PV systems, compared to the COSMO model predictions.

| Data | Measure | 2018 | | 2019 | |
|------|---------|------|------|------|------|
| | | Linear | Non-linear | Linear | Non-linear |
| **60min average** | **MBE** | -8.78 (-47.6) | -9.16 (-49.0) | 7.32 (59.3) | 8.06 (64.6) |
| | **RMSE** | 25.77 (139.7) | 26.06 (139.2) | 22.22 (180.1) | 23.52 (188.6) |

Figures 14 and 15 show the same results using measured pyranometer data to infer the COD. These retrievals show a similar trend as the PV-based ones: once again it is evident that the COD is mostly below 10, and in this range the retrieved data has a positive relative bias. There are several possible reasons for this: firstly it is evident from Eq. (11) that the retrieval is more sensitive to errors in inverted irradiance (transmission) for smaller COD. On the other hand, it must also be noted that the efficiency of both PV modules and silicon-based pyranometers shows a logarithmic dependence on irradiance, so that any inaccuracies in the PV model parameters or the pyranometer calibration will have a larger effect on the inverted irradiance under low light conditions (higher COD). In addition, since the LUT is constructed with water clouds, the effect of optically thin ice clouds is not properly taken into account. Since ice particles scatter slightly less in the forward direction, $1 - g \simeq 0.3$ is larger than for water clouds ($1 - g \simeq 0.15$) and thus a smaller optical depth could lead to similar irradiance at the ground [cf. Eq. (11)]. Thirdly, since clouds become more absorbing in the near infrared and considering that silicon PV is sensitive to wavelengths up to 1200 nm, spectral effects could also lead to a bias in the results. In general it must be said that even with measurements at two different wavelengths there exists an ambiguity in the determination of effective radius and COD (Nakajima and King, 1990), so that in the case of spectrally integrated PV-inverted irradiance one cannot expect to have enough information to accurately determine cloud optical properties in all situations.

Notwithstanding the bias in COD retrievals, the ground-based method presented here can still complement satellite retrievals, in particular due to the potentially higher spatiotemporal resolution achievable with large amounts of high frequency data spread over a large area. Once again, for the summer months the COSMO data show a large mean bias error of COD = 7.69, even for large values of COD, confirming the findings of Frank et al. (2018).

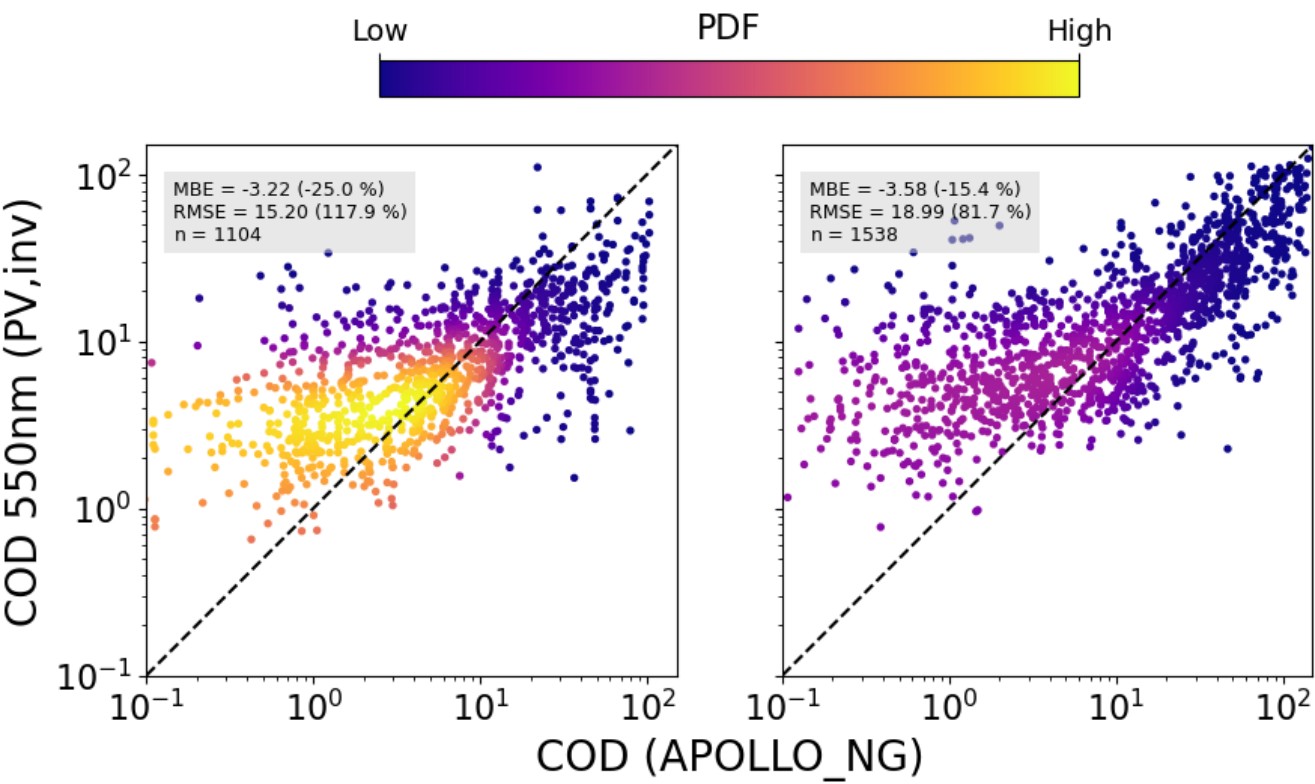

**Figure 12.** Combined comparison of COD retrieved from PV power measurements under completely cloudy conditions using the DISORT LUT with that from APOLLO_NG, for all stations and for 2018 (left) and 2019 (right), using the linear temperature model and averaged over 60 minutes.

## 5 Conclusions and Outlook

In summary, in this work a framework for extracting both global tilted and horizontal irradiance from PV power data has been presented and a first test for retrieving cloud optical depth is carried out. The algorithm makes use of state-of-the-art radiative transfer solvers in libRadtran, in conjunction with different sources of data for the state of the atmosphere, in particular the aerosol and water vapour content. The calibration procedure uses an explicit calculation of diffuse and direct irradiance, taking into account the spectral response of the relevant PV technology as well as the optical properties of the glass surface. Where

necessary the module temperature is modelled using weather model data for ambient temperature and wind speed input. The PV systems are calibrated using a libRadtran clear sky simulation with the DISORT solver, with inputs from the COSMO model and AERONET, and the algorithm can be adapted to each system and situation, depending on which parameters are known and which need to be determined by non-linear optimisation.

Once calibrated, the measured PV data is used to extracted global tilted irradiance under all sky conditions. In order to

645 take into account the spectral mismatch between the spectral response of PV modules and the entire broadband spectrum, a

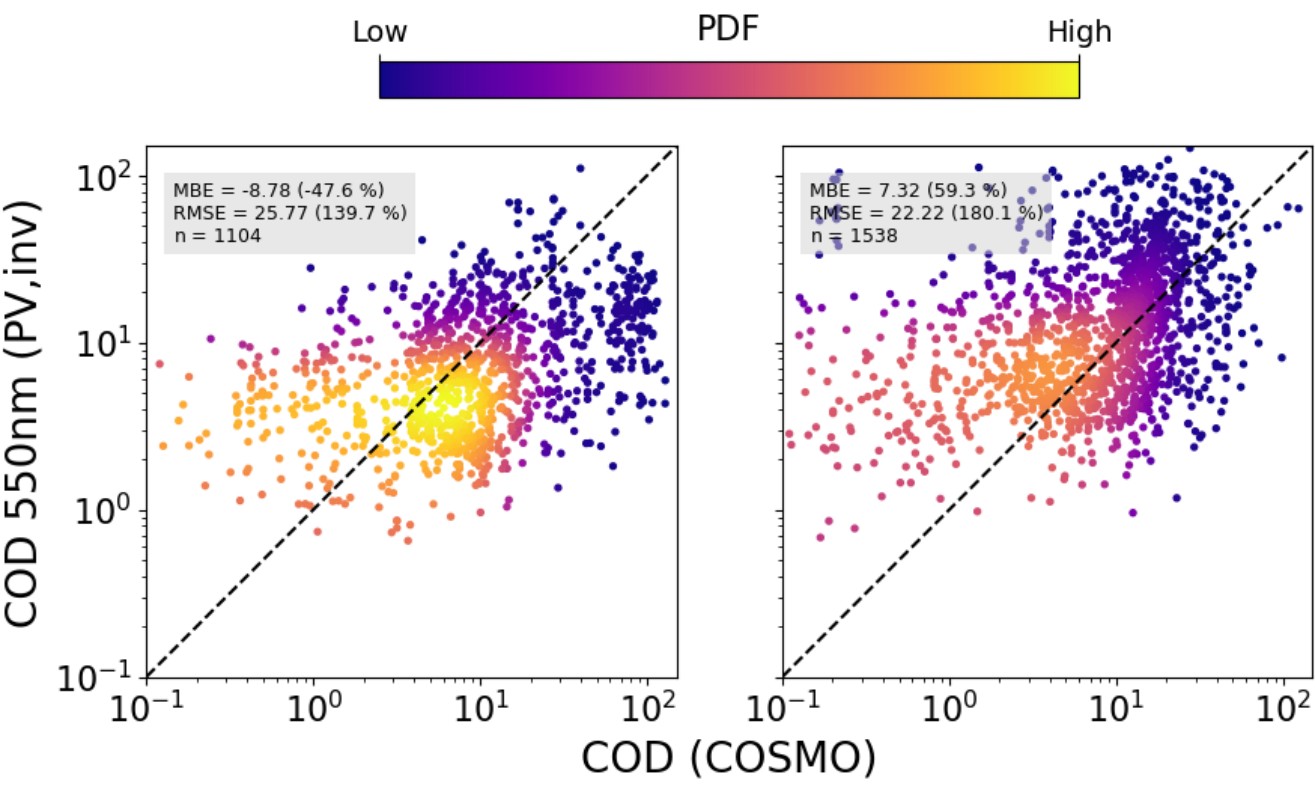

**Figure 13.** Same as 12, compared to the total COD from the COSMO model.

situation-dependent fit for the dependence of this mismatch on atmospheric conditions is performed, using simulated data for clear sky conditions and the water vapour and aerosol load of each site. The GTI is then compared to that measured by tilted pyranometers: the retrieved GTI at 1-minute (15-minute) resolution has a mean bias error of 18.68 W m$^{-2}$ (34.47 W m$^{-2}$) averaged over the two measurement campaigns, using the linear temperature model. The non-linear Faiman temperature model achieves a mean bias error of 8.77 W m$^{-2}$ for those systems with 1-minute data, but for those with 15-minute the calibration algorithm fails to find an optimal solution in several cases, so that in the end the mean bias error is 91.98 W m$^{-2}$. This shows that an accurate calibration is essential in order to accurately extract irradiance.

The inverted GTI is used to find the global horizontal irradiance using three different methods: (i) under persisting clear or (ii) cloudy conditions a lookup table based on a 1D DISORT simulation is used in order to find either the AOD or COD, and thus the global horizontal irradiance; (iii) under broken cloud conditions a LUT based on 3D MYSTIC simulations is used to translate the tilted to horizontal irradiance, using the geometry of the system, sun position and cloud fraction as inputs. The retrieved GHI is then compared to pyranometer measurements: in the case of the 1D LUT method, with 1-minute data under clear (cloudy) skies, the mean bias error is 13.79 W m$^{-2}$ (16.09 W m$^{-2}$) for the linear model and 5.79 W m$^{-2}$ (12.71 W m$^{-2}$) for the non-linear temperature model. Comparison of the 15-minute GHI retrievals with CAMS data reveals a positive bias

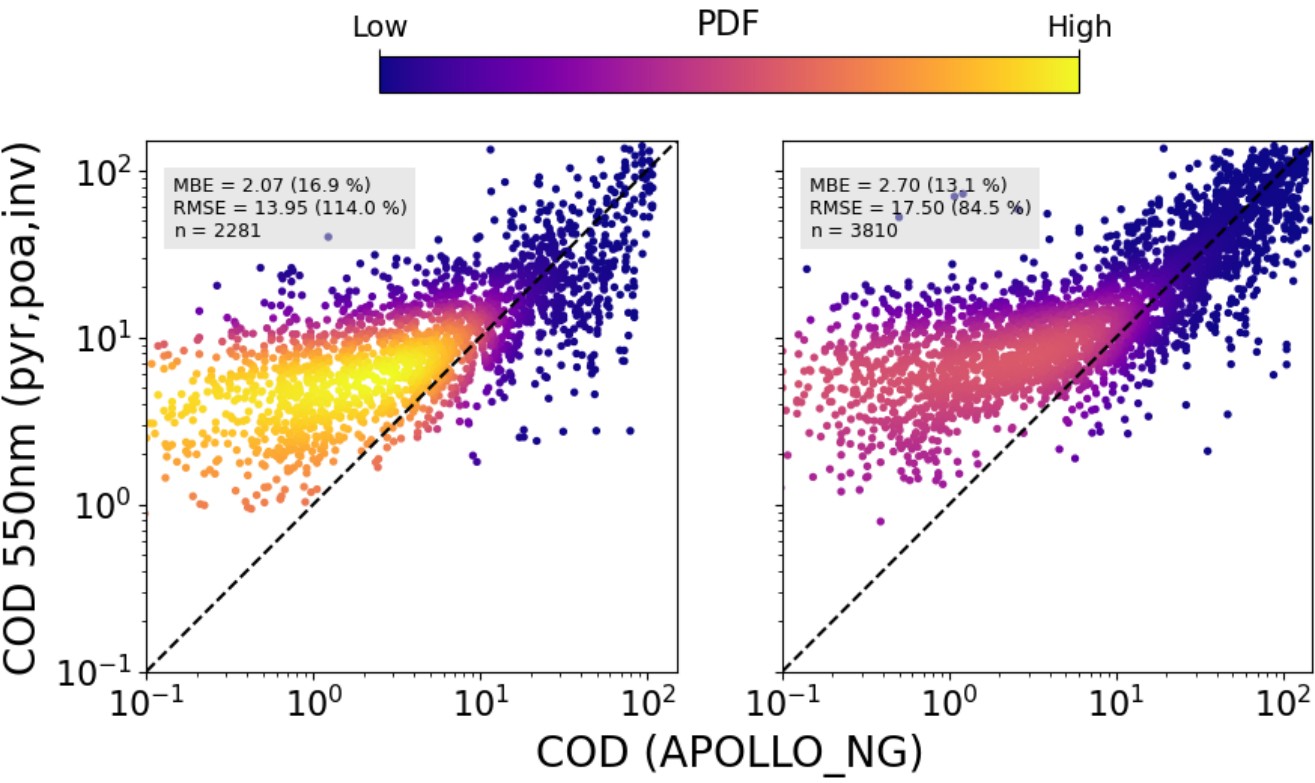

**Figure 14.** Combined comparison of COD retrieved from tilted (plane-of array = "poa") pyranometer measurements under completely cloudy conditions using the DISORT LUT with that from APOLLO_NG, for all stations and for 2018 (left) and 2019 (right), using the linear temperature model and averaged over 60 minutes.

under clear skies of 3.53 W m$^{-2}$ (2.27 W m$^{-2}$) for the linear (non-linear) model, whereas under cloudy skies there is a negative bias of -9.90 W m$^{-2}$ and -8.91 W m$^{-2}$ respectively. Considering the difference been point measurements and satellite pixels, as well as the inherent bias from considering only periods with 100% cloud fraction for the DISORT method, these results must be interpreted with care. In the case of clear skies, small errors in the temperature model can lead to large errors in extracted irradiance, and the case of cloudy skies, simplifying assumptions on cloud type can lead to errors in COD and extracted irradiance.

The retrieved GHI shows a large bias when comparing it with COSMO model data, thus confirming the results of Frank et al. (2018); Zubler et al. (2011): under clear skies the 60-minute averaged GHI has a mean bias error of 60.92 W m$^{-2}$, since COSMO has in general a too large aerosol load, whereas under cloudy skies in summer the MBE is -38.36 W m$^{-2}$, since under cloudy conditions the COSMO model generally tends to overestimates the irradiance. The latter result is also confirmed by the COD retrievals: in summer there is a positive bias of 7.69 for the retrieved COD relative to COSMO – the COD in the weather model is thus on average too small.

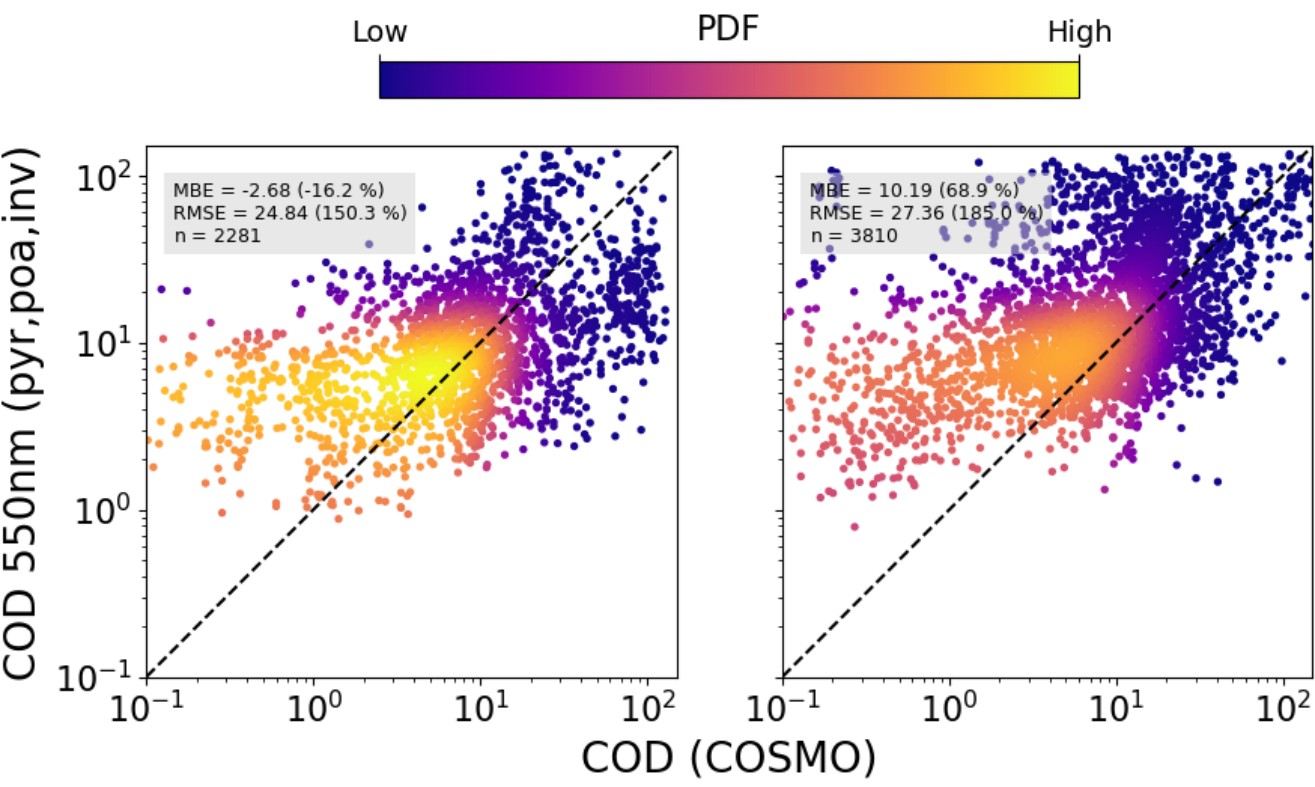

**Figure 15.** Same as 14, compared to the total COD from the COSMO model.

Overall, the largest source of error in the model chain comes from the PV model itself – an accurate calibration is vital in order to be able to extract irradiance reliably. In general the non-linear temperature model performs the best with high frequency PV data. In this regard it is helpful to use measured module temperature rather than relying on temperature models. More accurate results could also be achieved by using PV current measurements, since in this case the temperature dependence is almost negligible. This will be explored in future work.

The DISORT LUT is only employed during periods of persistent cloudy or clear sky conditions in order to infer COD or AOD, respectively, whereas the effect of 3D transport of photons is only taken into account with the MYSTIC LUT for the GHI. This means that the algorithm in its present form can only extract direct and diffuse components reliably under stable conditions for which 1D radiative transfer is still a reasonable approximation. A possible extension to this will be studied in future work, in which explicit 3D simulated cloud fields will be used in conjunction with the 1D DISORT simulation in order to quantify the error that results from neglecting 3D radiative transfer. It is only once the three dimensionality of atmospheric radiative transfer as well as additional information on cloud type is taken into account that one can accurately extract the direct and diffuse irradiance components under broken cloud conditions.

Another aspect not taken into account in this study is the possible gain in using several different PV systems that are close enough to each other so as to be able to see the same or similar portions of the sky. In this case it is conceivable that one could extract more information about cloud properties and irradiance compared to that obtained from just one system. Indeed, the rapid proliferation of PV installations could make such multi-sensor inversions an interesting future prospect.

The ultimate goal of this work is to show that PV power data can be used to infer global horizontal irradiance and optical 690 properties of the atmosphere, and the algorithm presented here is the first step in this direction. Although it is clear from the above that this is feasible, moving towards operational use would require several further steps. The largest source of bias comes from the calibration step, in particular the effect of temperature. Access to direct current (DC) data on the inverter level would allow a much more accurate extraction of the irradiance, without the confounding effect of temperature, since the dependence of MPP current on temperature is an order of magnitude smaller than that of MPP voltage and power. Since inverter data is 695 commonly available in industry, this should be possible provided one has the legal right to access the data. Additionally, the data pre-processing needs to be automated. For instance, one needs to exclude PV systems and/or data subsets that do not meet certain criteria such as shading of PV modules or inverter clipping, and one needs to be able to detect clear sky periods automatically, even if the system orientation is unknown. This could be done by developing a hybrid approach using both physical modelling and artificial intelligence (AI) algorithms for pattern recognition. In addition, the calibration procedure 700 itself could be augmented with AI, to find the unknown parameters more effectively.

Once these aspects are overcome and appropriate agreements with industry partners are made for access to the data, there should be nothing standing in the way of operational use. If this could be achieved at a large scale it would allow a better characterisation of solar irradiance at the ground and open up several possibilities for improving PV power forecasts at different time horizons. At shorter time (sub-hourly) scales one could use the additional information on irradiance variability as further 705 input to empirical forecasts based on statistical methods, whereas at longer time scales (more than three hours) these data could be assimilated into weather models. In order for this to make a difference one needs a much larger data set of PV systems for the analysis, which then requires further automation. First steps in this direction are currently being explored.

*Code and data availability.* Data is available for download from Zenodo (Barry et al., 2023), and code is available from Zenodo (Barry, 2023), linked to GitHub.

**Appendix A: Modelling details**

**A1 Parametric power model**

In Buchmann (2018) a simple model is proposed, based on the combination of several different modelling steps from the literature (see for instance Skoplaki and Palyvos (2009) or Dows and Gough (1995)). Here the PV power output is written as

$$P_{\text{AC,mod}} = A\,\eta_{\text{DCAC}}\,\eta_{\text{module}}(T, G^{\angle}_{\text{tot,SW},\tau}, v_{\text{wind}})\,G^{\angle}_{\text{tot,PV},\tau} = A\,\eta_{\text{DCAC}}\,\eta_{\text{module}}(T, G^{\angle}_{\text{tot,SW},\tau}, v_{\text{wind}})\big[G^{\angle}_{\text{dir,PV},\tau} + G^{\angle}_{\text{diff,PV},\tau}\big],$$

where $A$ is the surface area of the PV system, $\eta_{\mathrm{DCAC}}$ is the converter efficiency, and the direct and diffuse components of the irradiance in the plane-of-array and underneath the glass covering (see Sect. A2) are given by $G_{\mathrm{dir,PV},\tau}^{\angle}$ and $G_{\mathrm{diff,PV},\tau}^{\angle}$ respectively. The temperature-dependent module efficiency is defined by (Evans and Florschuetz, 1977)

$$\eta_{\mathrm{module}}(T, G_{\mathrm{tot,SW},\tau}^{\angle}, v_{\mathrm{wind}}) = \eta_{\mathrm{module,n}}\left[1 - \zeta(T_{\mathrm{module}} - T_{\mathrm{n}})\right], \tag{A2}$$

where $\eta_{\mathrm{module,n}}$ and $\zeta$ are the module efficiency and temperature coefficient at STC, i.e. at $T_{\mathrm{n}} = 25°\mathrm{C}$, $G_{\mathrm{tot,SW,n}}^{\angle} = 1000\ \mathrm{W\ m^{-2}}$
and air-mass of 1.5. Note that in principle one could include a logarithmic dependence of the module efficiency on irradiance (Sauer, 1994), which is however not considered here. The module temperature is modelled using both the linear model (Tamizh-Mani et al., 2003) defined by

$$T_{\mathrm{module}} = u_0 T_{\mathrm{ambient}} + u_1 G_{\mathrm{tot,SW},\tau}^{\angle} + u_2 v_{\mathrm{wind}} + u_3 T_{sky}, \tag{A3}$$

as well as the non-linear model (Barry et al., 2020; Faiman, 2008) defined by

725 $$T_{\mathrm{module}} = T_{\mathrm{ambient}} + \frac{G_{\mathrm{tot,SW},\tau}^{\angle}}{u_1 + u_2\, v_{\mathrm{wind}}} + u_3\left(T_{\mathrm{sky}} - T_{\mathrm{ambient}}\right), \tag{A4}$$

where $v_{\mathrm{wind}}$ is the wind speed at 10m above the ground, $T_{\mathrm{sky}}^4 = G_{\mathrm{LW}}^{\downarrow}/(\epsilon\,\sigma)$ defines the sky temperature and $u_{0,1,2,3}$ are model parameters. Here an emissivity of $\epsilon = 1$ is assumed and $\sigma$ is the Stefan-Boltzmann constant. Note that Eqs. (A1), (A2) and (A3) ((A4)) can be combined into the general non-linear expressions given in Eq. (1) (Eq. (2)) (see for instance Skoplaki and Palyvos (2009); Dows and Gough (1995)), where in this special case the coefficients are given by

730 $$b_1 = s\,(1 + \zeta\,25), \quad b_2 = -u_1 s\zeta, \quad b_3 = -u_0 s\zeta, \quad b_4 = -u_2 s\zeta, \quad b_5 = -s\zeta u_3 \tag{A5}$$

for the linear and

$$b_1' = s\,(1 + \zeta\,25), \quad b_2' = -\frac{u_1}{s\zeta}, \quad b_3' = s\zeta(u_3 - 1), \quad b_4' = -\frac{u_2}{s\zeta}, \quad b_5' = -s\zeta u_3. \tag{A6}$$

for the non-linear temperature model, where $s \equiv A\,\eta_{\mathrm{DCAC}}\,\eta_{\mathrm{module,STC}}$ is a constant scaling factor. The model equations in Eqs. (1) and (2) are used in this work, and the coefficients $u_{0,1,2,3}, \zeta, s$ are allowed to vary freely, unless their a priori values are
735 known from datasheets and/or from temperature modelling. In cases where measured temperature is available the parameters $u_{0,1,2,3}$ fall away.

## A2 Optical model of glass covering

### A2.1 Model formulation

In order to model the optics of the glass surface of the PV modules the following equation for the transmission of photons as a
740 function of the incident angle $\Theta$ is used (De Soto et al., 2006; Sjerps-Koomen et al., 1996)

$$\tau_{\mathrm{PV}}(\Theta) = \exp\left(-\frac{\kappa L}{\cos\Theta_{\mathrm{r}}}\right)\left(1 - \frac{1}{2}\left[\frac{\sin^2(\Theta_{\mathrm{r}} - \Theta)}{\sin^2(\Theta_{\mathrm{r}} + \Theta)} + \frac{\tan^2(\Theta_{\mathrm{r}} - \Theta)}{\tan^2(\Theta_{\mathrm{r}} + \Theta)}\right]\right), \tag{A7}$$

where $\Theta_r$ is the angle of refraction from Snell's law ($n \sin \Theta_r = \sin \Theta$), $n$ is the index of refraction of glass, $\kappa$ is the glazing extinction coefficient and $L$ is the glazing thickness. The incident angle $\Theta$ is the angle between the solar position vector and normal vector of the PV array, defined by

$$\cos \Theta = \cos \theta_0 \cos \theta + \sin \theta_0 \sin \theta \cos(\phi_0 - \phi), \tag{A8}$$

where $\theta$ is the angle of inclination of the PV array, $\phi$ is its orientation, $\theta_0$ is the solar zenith angle and $\phi_0$ is the solar azimuth.

In principle one should take into account the wavelength dependence of $n$ and $\kappa$, however for most materials they are relatively constant, with $n$ increasing slightly at lower wavelengths, and since in practice the exact properties of the glass covering for each system are unknown it suffices to use the effective values for all wavelengths.

The so-called incidence angle modifier (see also Duffie and Beckman (2013)) is defined by the ratio of the transmission $\tau_{\mathrm{PV}}(\Theta)$ and the transmission at normal incidence, i.e.,

$$\tau_{\mathrm{PV,rel}}(\Theta) \equiv \frac{\tau_{\mathrm{PV}}(\Theta)}{\tau_{\mathrm{PV}}(0)}, \tag{A9}$$

where

$$\tau_{\mathrm{PV}}(0) = e^{-\kappa L} \left[ 1 - \left( \frac{n-1}{n+1} \right)^2 \right] \tag{A10}$$

$$= e^{-\kappa L} \frac{4n}{(n+1)^2}. \tag{A11}$$

The use of a relative transmission coefficient is justified from the fact that the absolute transmittance is already taken into account when characterising the solar cell under standard conditions (Sjerps-Koomen et al., 1996). The normalisation with $\tau_{\mathrm{PV}}(0)$ means that the result is less sensitive to the product $\kappa L$ and more on the angle of incidence. In the forward model the wavelength-integrated direct irradiance as well as the diffuse radiance beams are each multiplied with the factor $\tau_{\mathrm{PV,rel}}(\Theta)$ in order to take into account the attenuation due to the glass surface. The values of the extinction coefficient and thickness of the glass are fixed to (De Soto et al., 2006) $\kappa = 4$ m$^{-1}$ and $L = 0.002$ m, respectively, whereas the effective refractive index $n$ is allowed to vary ($n = 1.526$ with an a-priori error of 1%). In principle one could also vary the product $\kappa L$ that controls the absorption, but as mentioned above the incident angle modifier approach is applicable for a wide range of glass covers (Duffie and Beckman, 2013; Klein, 1979).

## A2.2 Optical model in forward model calibration

In the forward model calibration, the transmission function can then be used to calculate the direct and diffuse irradiance, $G^{\angle}_{\mathrm{dir,PV},\tau}$ and $G^{\angle}_{\mathrm{diff,PV},\tau}$ as

$$G^{\angle}_{\mathrm{dir,PV},\tau} = \frac{\cos \Theta}{\cos \theta_0} G^{\downarrow}_{\mathrm{dir}} \tau_{\mathrm{PV,rel}}(\Theta), \tag{A12}$$

and

$$G^{\angle}_{\mathrm{diff,PV},\tau} = \int_{\phi}^{2\pi+\phi} \int_{-\theta}^{\pi/2-\theta} L^{\Omega}_{\mathrm{diff}} \cos \Theta' \, \tau_{\mathrm{PV,rel}}(\Theta') \, d\Omega, \quad \text{for } \cos \Theta' \geq 0, \tag{A13}$$

where $L_{\text{diff}}^{\Omega}$ is the diffuse radiance distribution calculated by DISORT. In this way there is no need for an empirical incidence angle modifier, since the direction of each diffuse photon is explicitly described. The same formulation is used to calculate $G_{\text{dir,SW},\tau}^{\angle}$ and $G_{\text{diff,SW},\tau}^{\angle}$, i.e. over the whole wavelength range.

### A2.3  Inversion of the optical model

For the inversion of the PV model, two different methods are used: for the extraction of cloud optical depth with DISORT the optical model can be explicitly taken into account in the radiative transfer simulation, whereas for the direct extraction of GTI and its translation to GHI with the MYSTIC lookup table the empirical formulation (Duffie and Beckman, 2013) for the effective angle for diffuse photons as a function of tilt angle $\theta$,

$$\Theta_{\text{diff}} = 59.7 - 0.1388\,\theta + 0.001497\,\theta^2 \tag{A14}$$

is used for all time points with clearness index below 0.3, so that the final inverted GTI is given by

$$G_{\text{tot,SW,inv}}^{\angle} = \frac{G_{\text{tot,SW,inv},\tau}^{\angle}}{\tau_{\text{PV,rel,eff}}}, \tag{A15}$$

where

$$\tau_{\text{PV,rel,eff}} = \begin{cases} \tau_{\text{PV,rel}}(\Theta_{\text{dir}}) & \text{if } k_i \geq 0.3 \\ \tau_{\text{PV,rel}}(\Theta_{\text{diff}}) & \text{if } k_i < 0.3 \end{cases} \tag{A16}$$

is the effective incidence angle modifier for clearness index $k_i$.

### A3  Spectral mismatch fitting procedure

The ratio of silicon PV irradiance to broadband irradiance as defined in Eq. (5) is a function of atmospheric composition (primarily water vapour content) and angle of incidence of the incoming solar beam. Indeed, the shape of the diurnal variation in $\xi_{\text{spec,GTI}}$ is dependent on the azimuth angle of the PV plant, as shown in Figs. A1 and A2, where the ratio is plotted for both GHI and GTI, for two different PV systems with different orientation, using libRadtran clear sky simulations from the
different measurement campaigns. The points in the upper (lower) plots in each figure are coloured according to precipitable water (AOD), and it is evident that for a given incident angle the water vapour plays the largest role in determing the ratio.

The mismatch ratio remains at roughly 0.83 throughout the day, depending on water vapour column. In the case of PV12 in Fig. A1, one can see that in the mornings the ratio is smaller, i.e. more red light, whereas in the evenings the ratio is larger, since the panel is facing more to the south east. This means that the panel detects more diffuse light in the evenings. The behaviour
at PV15 (Fig. A1) is the opposite – in the morning the ratio is larger, since the panel looks more to the south west (mornings higher diffuse component, i.e. more blue light), whereas in the evenings it is smaller. However in the summer time when the sun goes down far to the north, the diffuse component again plays a role, so that the ratio becomes larger (right hand plot).

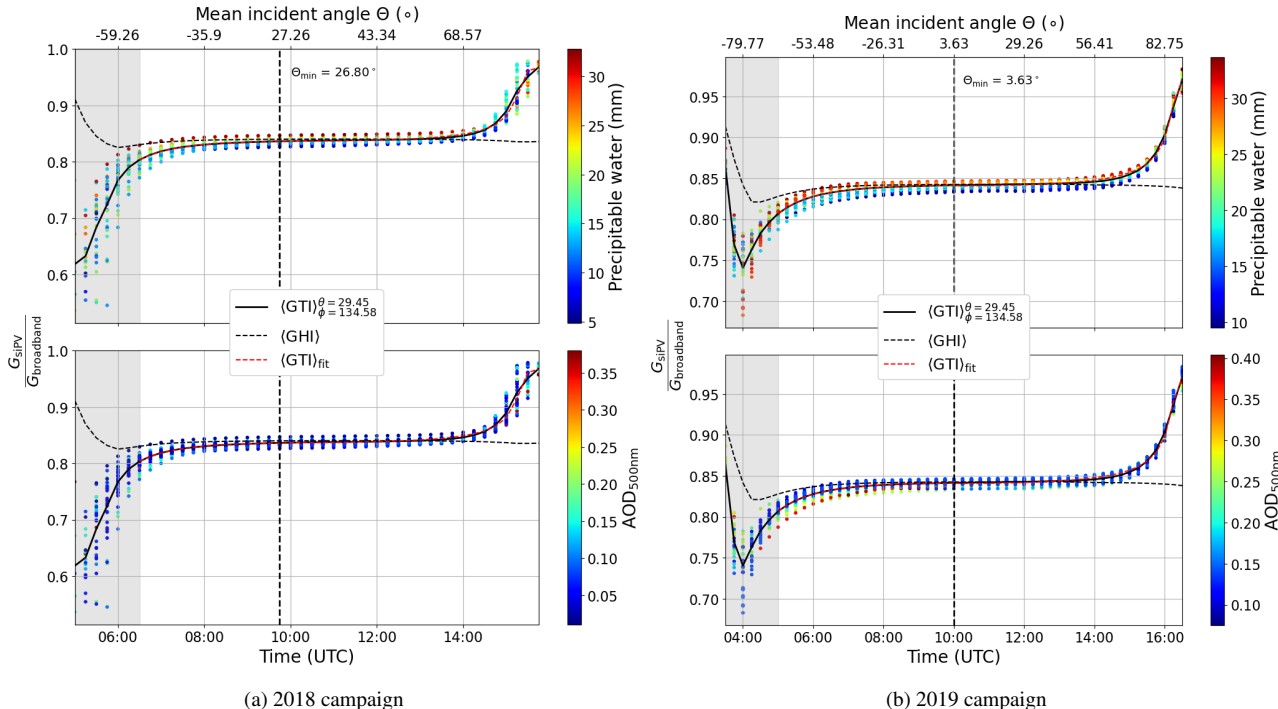

**Figure A1.** Spectral mismatch ratio at PV12 (azimuth angle $\phi \simeq 135°$) as a function of time, mean incident angle as well as water vapour column (upper plots) and AOD (lower plots), both (a) 2018 and (b) 2019 measurement campaigns. $\Theta_{\mathrm{min}}$ is the minimum incident angle.

As shown in the Figures above, the data is first grouped by time of day in order to calculate the mean value, then split into two halves on either side of the minimum incident angle ($\Theta_{\mathrm{min}}$). Each branch can then be fitted with the function

$$\langle \xi_{\mathrm{spec,GTI}} \rangle = x_0 \exp\left( -\frac{x_1}{\cos \Theta} - \frac{x_2}{\cos \Theta^2} \right), \tag{A17}$$

shown as the red dashed line in Figs. A1 and A2. This shows the general form of the diurnal variation in spectral mismatch, and that for silicon PV the ratio is about 83% for most of the day.

In order to capture the effect of precipitable water and AOD, the function

$$\xi_{\mathrm{spec,GTI}} = p_0 \exp\left( -p_1 w - p_2 a \right) \tag{A18}$$

is fitted to the irradiance data for each time step (averaged over all days of the respective measurement campaign), where $w$ is the precipitable water from COSMO and $a$ the AOD at 500nm from AERONET. The fit coefficients are then interpolated over the entire dataset in order to calculate $\xi_{\mathrm{spec,GTI}}$ at any time of day. Future work will examine the effects of clouds on the spectral mismatch factor – here the clear sky fit is applied to all data, which due to whitening of the skylight by clouds could lead to a bias in the final result.

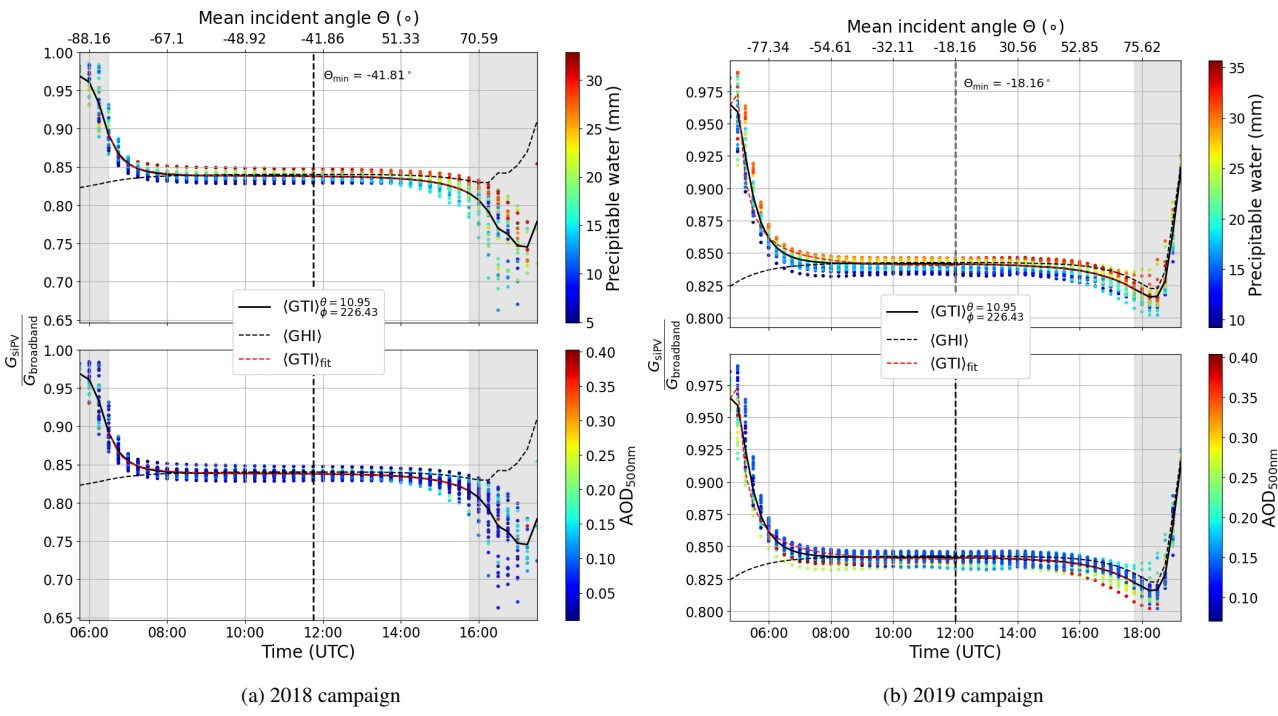

(a) 2018 campaign

(b) 2019 campaign

**Figure A2.** Same as A1, for PV15 (azimuth angle $\phi \simeq 226°$)

*Author contributions.* The two measurement campaigns were designed and coordinated with contributions from all authors, and the installation and calibration of the various measurement devices was performed by NK, CS, RY, HD, JW, FG, PH and MS. The PV model was conceptualised by JB, TB, KP, AH-C and SM; the software and simulations to implement and validate the model were developed and carried out by JB, with contributions from JG, FG and AH-C. FG wrote several Python subroutines in order to import weather model data and also developed the MYSTIC lookup table. CE and BM introduced several customised features into libRadtran for this work; LS and MS-H provided weather model and satellite data. JB prepared the manuscript with contributions from all co-authors.

*Competing interests.* One author (BM) is a member of the editorial board of AMT. The peer-review process was guided by an independent editor, and the authors have also no other competing interests to declare.

*Acknowledgements.* The authors acknowledge support from the German Federal Ministry for Economic Affairs and Energy (BMWi) under the project "MetPVNet: Entwicklung innovativer satellitengestützter Methoden zur verbesserten PV-Ertragsvorhersage auf verschiedenen Zeitskalen für Anwendungen auf Verteilnetzebene", with project code 0350009. The DWD is thanked for the provision of COSMO model data. JB would like to thank Lukas Tirpitz for helpful comments and suggestions in the development of this work. In addition, the authors thank the thorough review work of two anonymous referees, whose efforts significantly contributed to improving the manuscript.

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
