# Peer review of "Irradiance and cloud optical properties from solar photovoltaic systems"

_Atmospheric Measurement Techniques, 2022_

## Author Response (AR1)

**Our responses to the comments of the reviewers for manuscript amt-2022-335**

9. August 2023, James Barry et al.

First referee (**https://doi.org/10.5194/amt-2022-335-RC1**)

*This paper presents a method to calculate the global irradiance at the surface and some atmospheric properties using photovoltaic panels. To achieve this, the inversion algorithm accommodates both a description of the hardware and one of the characteristic state of the atmosphere.*

*I found the article very well written, clear in exposition and scientifically correct in its premises, logical development and description of results. But at the same time it is by no means easy to digest. The text as a whole is very complex, with constant cross-references between one section and the next, and I am not sure it is easily approached by the average reader. This is due to the need to describe all those components of the experiment that determine, at the end of the chain, the final accuracy of the inversion.*

*In another situation, with another article, I would have requested the authors to streamline the text and make it easier for a wider audience to understand. In this case, however, it is clear to me that this is not easy, given the experimental complexity of the task the authors are facing.*

*So, as far as I am concerned, I am asking for minor revisions. These are mostly non-critical changes but should add some additional detail regarding the setup of the simulations and the reach and the description of the results.*

*General considerations after reading the full text*

1. *It is surprising that the mean bias error increases with wider temporal aggregations. Can the authors comment on this?*

Our response: Indeed, one would expect the bias to increase with coarser time resolution. In our case, the opposite effect is seen, which is due to the bias being dominated by errors in the calibration procedure, as well as the higher data quality for the systems with high frequency measurements. Even though the calibration itself is always performed with 15-minute data under clear sky conditions, the calibration algorithm performs better when taking high-resolution data that has been aggregated to 15 minutes. One reason for this is simply that the AÜW measurements of PV power every 15 minutes are based on energy meter readings for the last 15 minutes, which were shifted by 7.5 minutes, as described in section 3.1. This was done in order to bring different data sources with different aggregation procedures into alignment. Using the 1 Hz data leads to better 15-minute averages, and thus more accurate calibration.

Another reason could be that certain outliers in the 15-minute data should have been excluded from the inversion (see the points forming horizontal lines in Figure 6). We checked this and found that this would improve the RMSE, but not the bias, which indeed comes from certain systems with poor calibration. For completeness, these erroneous values are excluded, and the results recalculated.

After re-running the analysis we decided to separate clear and cloudy skies more explicitly, which will be discussed below. It will be seen that in some cases under clear skies the 15-minute data does show reduced error, but not always. This again points towards both better data quality for the 1 Hz measurements as well as more accurate calibration.

In addition, we discovered an error in the GTI statistics – the cosine bias correction had not been correctly applied to the pyranometer data. After applying this the bias error in GTI with respect to pyranometers reduces significantly.

Changes to the manuscript:

- Erroneous data that had been originally included was removed from the datasets, and the results were recalculated. This changes the RMSE and bias error in all tables.
- Cosine bias correction was applied to the TROPOS pyranometer data, thus reducing the bias error of the inverted GTI from PV systems.

2. *I assume that the efficiency of the photovoltaic modules is highest in clear sky conditions and lowest in overcast conditions. If correct, how does the lower efficiency relate to the accuracy of the derived cloud optical light extinction? Is it a linear relationship?*

Our response:  We fully appreciate the question! The efficiency of a solar cell is a function of both irradiance and temperature, and can be shown to be logarithmically proportional to irradiance and linearly proportional to temperature.  In that sense any inaccuracies in the PV model parameters will have a larger effect in the inverted irradiance under low light conditions. Indeed, this is the main source of error in the retrievals – the more parameters used to model the PV power the greater the error in the inverted irradiance.

Changes to the manuscript:

- In the appendix in line 607 the following was added:

Note that in principle one could include a logarithmic dependence of the module efficiency on irradiance (Sauer, 1994), which was however not considered here.

- In the discussion in line 523, the following was added:

On the other hand, it must also be noted that the efficiency of both PV modules and silicon-based pyranometers shows a logarithmic dependence on irradiance, so that any inaccuracies in the PV model parameters or the pyranometer calibration will have a larger effect in the inverted irradiance under low light conditions (higher COD).

3. *COD_PV is verified (at least in my eyes it is not a validation) with the values of COSMO and APOLLO_NG. But if I am not mistaken, COSMO and APOLLO_NG are at the same time the source of some critical inversion inputs. In this sense it is more of a verification than a validation. Since there are now established methods for determining even the optical thickness of clouds from ground-based measurements (e.g.,* www.atmos-meas-tech.net/8/1361/2015/)*, I wonder if it is not worth trying to find some stations in the area of interest that provide true validation as independent.*

Our response: That is partly correct, the inverted COD is not "validated" with COSMO and APOLLO_NG but rather simply compared to the COD from those sources. APOLLO_NG is however not used as input, but COSMO is, specifically the atmospheric composition from COSMO is used for the radiative transfer simulations.

Indeed, it would be more appropriate to validate the PV-based COD with another ground-based measurement, however to our knowledge there were no appropriate meteorological stations in the area that could have been used for this purpose. Although there are several DWD stations in the area (in Kempten, Oberstdorf and Hohenpeissenberg), these provide information on irradiance (direct and diffuse), but not on cloud optical properties (see Becker and Behrens, (2012)). Thus, a true validation would have to be done for another dataset with PV systems closer to a measurement station that has ground-based retrievals of COD. This will be done in future field experiments.

Changes to the manuscript:

- A paragraph was added at the end of section 3.1:

In order to validate the PV- and pyranometer-based COD retrievals, it would be appropriate to use another ground-based source of cloud optical properties, however unfortunately there were no appropriate meteorological stations in the immediate area that could have been used for this purpose. Although there are several DWD stations in the Allgäu region (in Kempten, Oberstdorf and Hohenpeissenberg), these provide information on irradiance (direct and diffuse), but not on cloud optical properties (see Becker and Behrens, (2012). Thus, a true validation would have to be done for another dataset with PV systems closer to a measurement station that has ground-based retrievals of COD. For this reason, the COD retrievals were simply compared to the corresponding cloud properties from weather models and satellite data.

*Specific comments*

*Abstract, L 7: it is stated that results on COD and AOD will be presented. While several scatterplots are presented and discussed for COD in the text, I found no mention of any retrieval of AOD. One might guess that the results with COD<2 contain, within them, also those of AOD. If true, then the text of the abstract is misleading, as one would expect a retrieval of atmospheric opacity directly due to aerosols. This would probably explain why, for the generation of LUTs with MYSTIC, there is no additional dimension referring to aerosols.*

Our response: We agree that this could be seen as misleading. In this paper, the retrieval of AOD was not explicitly discussed, since the results obtained with this method show large biases due calibration error, coupled with the very small AOD in the Allgäu region (as discussed in the paragraph starting in L333). However, the retrievals of GHI under clear sky conditions implicitly contain AOD, since the optical depth is used in this case as a proxy to "reverse transpose" the tilted irradiance to horizontal irradiance. Perhaps this was not clearly communicated. If one looks at the schematic diagram in Figure 2, in order to retrieve GHI under clear sky conditions the algorithm first finds the AOD and then uses the 2D LUT to retrieve the direct and diffuse components, adding up to the GHI.

In order to make this distinction more explicit we decided to completely change the way the GHI results are presented: the clear sky days were separated from the cloudy days in the statistics. This allows one to evaluate the accuracy of the difference lookup table methods: (i) DISORT AOD LUT under clear skies, (ii) DISORT COD LUT for completely clouds periods and (iii) MYSTIC 3D GHI LUT under broken clouds. The statistics are fundamentally different and one can now see the limitations of each method clearly. The discussion was thus updated appropriately.

Changes to the manuscript:

- Figure 2 was updated, firstly to make the distinction between clear/overcast vs. broken clouds clearer.

- In the abstract, the following sentence was deleted: Specifically, the aerosol (cloud) optical depth is inferred during clear sky (completely overcast) conditions.
- In section 2.3.4, the AOD was described as a proxy to get to GHI under clear sky conditions, with the following sentences:

Using the AOD-GTI lookup table, the AOD can be extracted on clear sky days, and from this also the direct and diffuse irradiance as well as the global horizontal irradiance. In this way the AOD is used as an intermediate step for the reverse transposition of GTI to GHI.

- In section 2.3.3, a sentence was added:

Note that the COD LUT also implicitly contains aerosol information as an input, since here the aerosol properties are fixed using the OPAC database (Hess et al., 1998), and the Ångström parameters from AERONET are used.

- All GHI results were separated according to the method used, changing the statistics. We decided not to calculate 60 minute averages (except for the comparison with COSMO data), since this leads to inaccurate results under broken cloud conditions due to the inherent limitations of the DISORT LUT method for COD.

*Introduction, L50 and ff: Can the authors add a sentence or two summarizing the accuracy of the cited methods?*

Our response. The method described in Engerer and Mills (2014) use a PV-based clear sky index and can achieve an MBE of 1.09% and a mean absolute percentage error (MAPE) of 4.1% for the PV power output under clear sky conditions, but the accuracy diminishes for partly cloudy skies, as expected (reaching up to 50% MAPE). Killinger et al. (2016) achieve an MBE of between -3.9% and -9.8% for the GHI, depending on the empirical model used for irradiance transposition. Marion and Smith (2017) achieve an MBE for the GHI of within ±1.5% using south-facing PV modules at 10°, 25° and 40° tilt angles. Halilovic et al. (2019) use an analytical method for the reverse transposition of GTI to GHI and achieve an MBE of between 0.1% and 2.1% for the resulting GHI, using data from silicon reference cell measurements at different tile and azimuth angles.

Changes to the manuscript:

- the introduction was updated with the following sentences:

Engerer and Mills, (2014) achieve a mean bias error of 1.09% for the PV power output under clear sky conditions, but the accuracy diminishes for partly cloudy skies, as expected. Killinger et al. (2016) achieve a mean bias error between -3.9% and -9.8% for the GHI, depending on the empirical model used for irradiance transposition, Marion and Smith (2017) achieve a mean bias error for the GHI of within ±1.5% using south-facing PV modules at 10°, 25° and 40° tilt angles. A similar approach is taken in Elsinga et al., (2017), in this case using a single diode PV model and a different decomposition model. In Halilovic et al., (2019) the authors replaced the original iterative approach used in Killinger et al. (2016) by an analytical method, to minimise computational cost, and achieve a mean bias error of between 0.1% and 2.1% for the resulting GHI, using data from silicon reference cell measurements at different tile and azimuth angles.

*P3 L73: what is the reasoning behind the 5 Wm-2 desired threshold?*

Our response: as discussed in the second paragraph of the introduction (L31-L49), state-of-the art satellite data has a mean bias of the order of 5 $Wm^{-2}$. In addition, this level of accuracy corresponds to the target accuracy for global radiation measurements as part of the Baseline Surface Radiation

Network, which was established within the frame of the World Climate Research Programme (McArthur, 2005). In that sense it would be desirable to reach this level of accuracy. However, even if this is not achieved, ground-based irradiance measurements and/or retrievals have the added advantage of superior spatiotemporal resolution, if the necessary data is available.

Changes to the manuscript:

- The following was added to the introduction:

In order for a PV-based determination of solar irradiance to viably complement the global coverage of state-of-the-art satellite measurements, a mean bias error of the order of 5 Wm$^{-2}$ would be desirable (see the discussion on CAMS and other satellite-based products above). This level of accuracy also corresponds to the target accuracy for global radiation measurements from the BSRN (McArthur, 2005). However, even if this is not achieved, ground-based irradiance measurements and/or retrievals can be seen as complementary since they have the added advantage of superior spatiotemporal resolution. The first step to achieve this is to accurately model…

*P5 L139-142: "It can be shown using the diode model [see for instance Sauer (1994); Abe et al. (2020)] that the maximum power point (MPP) current generated by a PV module is linearly dependent on the incident irradiance, and only very weakly dependent on temperature."*

*Surprising. But still no info on the nature of the MPP-irradiance under overcast skies.*

Our response: It is important here to distinguish between current and voltage. In this work the power was used as the data source for inversion, which leads to the added complication that although MPP current is linearly proportional to irradiance, the MPP voltage is logarithmically proportional to irradiance, so that the overall efficiency also shows such a logarithmic dependence. The MPP power is of course strongly dependent on temperature, but this comes from the voltage, not the current – the latter has only a very weak dependence on temperature.

Changes to the manuscript: this is taken care of when discussing the COD retrieval under overcast skies.

*Eq.1 and 2: even if obvious, \$tau is not introduced in the text. This also leaves the room open for me to ask how \$tau influences Eq.1 and 2. Is \$tau role linear on non-linear?*

Our response: This was in fact introduced in line 164 in the text: $\tau$ is simply a subscript indicating that an optical model has been employed to account for absorption and reflection by the glass surface of the PV module. Its effect is largely dependent on the angle of incidence, as discussed in Appendix A2. The final inverted GHI to be compared with other sources is then defined in Eq.8 as the irradiance "above" the glass surface, i.e., the optical model must also be taken into account in the inversion.

Changes to the manuscript:

- The introduction to $\tau$ was moved to directly under Equations 1 and 2, where the following was added:

Note that the subscript ``PV'' for the tilted irradiance $G_\mathrm{tot,PV,\tau}^{\angle}$ refers to the fact that only the relevant wavelength (in this case 300 nm to about 1200 nm for silicon PV modules) is considered, and the subscript ``$\tau$'' indicates that transmission through the glass surface of the PV panels has been taken into account with an optical model. Further details of the model employed here are given in Sections A1 and A2 in the Appendix, and the refractive index $n$ of the glass covering

is one of the parameters varied in the optimisation procedure. The subscript ``SW'' refers to all incoming shortwave photons -- the dependence of the spectral mismatch between the PV and SW irradiance bands on atmospheric water vapour and other factors will be discussed in Section 2.3.

*How about a direct aerosol radiative effect at play? Is this embedded in the simulations based on the OPAC database?*

Our response: Aerosols are considered in different parts of the model chain. For the forward model calibration (PVCAL) under clear sky conditions, the AERONET data from the Hochschule Kempten is used to find the Ångström parameters, which are then combined with the OPAC data base (using the continental average setting) to obtain an accurate irradiance proxy for calibration. For the inversion, the AERONET data and corresponding Ångström parameters are not used, however the OPAC setting remains. This means that there is implicit aerosol information on aerosol type in both the AOD and COD lookup tables.

Changes to the manuscript:

- In section 2.3.3, a sentence was added:

Note that the COD LUT also implicitly contains aerosol information as an input, since here the aerosol properties are fixed using the OPAC database (Hess et al., 1998), and the Ångström parameters from AERONET are used.

P14 L 330 : AOD = 0.01 and AOD = 1. What is the typical AOD for dust events reaching Central Europe?

Our response: The typical dust event reaching Europe does not have such a high AOD, but is rather characterised by small values of the Ångström exponent less than 1, indicating the presence of coarser dust particles. For example, one study of the climatology of dust events found a mean AOD of 0.155, 0.32 and 0.122 for dust plumes in southern, central and northern Europe (Mandija et al., 2018). Indeed, during the time between September 2018 and July 2019, the AOD at 500nm measured by the AERONET station on the rooftop of the Hochschule Kempten did not exceed 0.5. Just before the second measurement campaign, on the 25[th] and 26[th] of June 2019, a dust event hit Europe. In this case the loglinear fit from AERONET data shows an Ångström exponent of around 0.6, indicating the presence of dust particles. However, the AOD on these days was only about 0.23 at 500nm.

Changes in the manuscript:

- The following sentences were added:

In this context it must be noted that the typical dust event reaching Europe does not have such a high AOD, but is rather characterised by small values of the Ångström exponent less than 1, indicating the presence of coarser dust particles. For example, one study of the climatology of dust events found a mean AOD of 0.155, 0.32 and 0.122 for dust plumes in southern, central and northern Europe (Mandija et al., 2018).

- In addition, the text in parentheses in L333 was changed to

(during the measurement campaigns it did not exceed 0.5 at 550nm)

*P14 Section 2.3.5: I have two questions with respect to the setup of MYSTIC and its range of application.*

*(1) why wasn't MYSTIC also equipped with an additional dimension of AOD? I would expect that, especially in intermediate situations of broken cloudiness, the presence of aerosols may not only change*

*the intensity of irradiance but also its spectral behaviour. The latter is precisely identified by the authors as one of the factors contributing to the inversion uncertainty.*

Our response: Firstly, one must understand that the MYSTIC lookup table (based on 3D simulations) was simply used to convert the inverted tilted irradiance (GTI) to horizontal irradiance (GHI) under broken cloud conditions. Extensive sensitivity studies were carried out to understand the dependence of the relationship between tilted and horizontal irradiance on different atmospheric parameters such as AOD, albedo, liquid water content, etc. It turns out that AOD has no significant influence on this relationship, and that the most important parameters are the system geometry, sun position and the cloud fraction. The AOD was used in the context of the DISORT-based lookup table under clear sky conditions, in order to extract the GHI.

*(2) Later on in the text, Table 9 shows the limits of applicability of MYSTIC, but in the main text there is no discussion on the reasons why exactly those values are reported.*

Our response: There are three major reasons for the choice of limits of the MYSTIC LUT. Firstly, despite several optimisations like the reduction of the number of photons used for the Monte Carlo simulations, the computational demand for calculating the LUT is high. For this reason, 20 degrees was chosen as the lower limit for the SZA, since in the latitudes under investigation no smaller values occur. Similar considerations apply to the tilt angle of PV panels – in Allgäu, Germany one rarely encounters title angles larger than 50 degrees.

The second limiting factor relates to the derivation of a cloud mask and cloud fraction from the radiation measurements (see Eq. 12 in the text). Firstly, this is only possible when there is direct line of sight between the sun and the module or sensor, which limits the relative azimuth angle between the sun and the PV panel. Secondly, the derivation of a cloud fraction from temporally resolved radiation measurements becomes unprecise at large SZAs for geometrical reasons, so that the upper limit of the SZA was set to 60 degrees.

Finally, the cloud fraction limits are determined by two factors: Firstly, the LUT model was developed and tested for partly cloudy situations. The special cases of 0% and 100% cloud fraction were considered separately with the DISORT-based LUTs, as other parameters like AOD and COD become relevant here. Secondly, the exact cloud fraction limits (0.13 and 0.82) given in table 9 are constrained by the available cloud scenes from LES simulations.

Changes to the manuscript:

- This section was moved into Section 2.3.5, and then just referred to in Section 4.3.
- The following text was added to the manuscript:

Table 5 shows the limits of applicability of the MYSTIC LUT, for which there are three major reasons. Firstly, despite several optimisations like the reduction of the number of photons used for the Monte Carlo simulations, the computational demand for calculating the LUT is high. For this reason, 20° was chosen as the lower limit for the SZA, since in the latitudes under investigation no smaller values occur. Similar considerations apply to the tilt angle of PV panels – in Allgäu, Germany one rarely encounters title angles larger than 50°. The second limiting factor relates to the derivation of a cloud mask and cloud fraction from the radiation measurements [see Eq. (12)]. Firstly, this is only possible when there is direct line of sight between the sun and the module or sensor, which limits the relative azimuth angle between the sun and the PV panel. Secondly, the derivation of cloud fraction from temporally resolved radiation measurements becomes unprecise at large SZAs for geometrical reasons, so that the upper limit of the SZA was set to 60°. Finally, the cloud fraction limits are determined by two factors: firstly, the LUT model was developed and tested for partly cloudy situations. The special cases of 0% and 100%

cloud fraction were considered separately with the DISORT-based LUTs, as other parameters like AOD and COD become relevant here. Secondly, the exact cloud fraction limits (0.13 and 0.82) given in Table 5 are constrained by the available cloud scenes from LES simulations.

*P17 L 392: which overlap scheme for clouds? Can the authors be more specific on this point?*

Our response: For comparison with the cloud optical depths extracted from the PV systems, a two-dimensional COD field must be computed from the three-dimensional cloud variables generated by the COSMO model. For each grid cell, a cloud fraction variable in COSMO indicates which fraction of the cell is covered by clouds. To derive a vertically integrated COD we need to make an assumption as to how these clouds overlap in a model column. Following Scheck et al. (2018), we adopt the often used random-maximum cloud overlap assumption and employ the method of Matricardi (2005) to compute for each model column the vertically integrated COD for a number of subcolumns. From these subcolumn values we derive a mean COD for the cloudy part of the column. A total COD is then computed as the average of the column mean COD over 5 by 5 columns centred around the column containing the relevant ground station.

Changes to the manuscript:

- The following is added to the text:

For comparison with the cloud optical depths extracted from the PV systems, a two-dimensional COD field must be computed from the three-dimensional cloud variables generated by the COSMO model. For each grid cell, a cloud fraction variable in COSMO indicates which fraction of the cell is covered by clouds. To derive a vertically integrated COD, an assumption needs to be made as to how these clouds overlap in a model column. Following Scheck et al., (2018), the commonly used random-maximum cloud overlap assumption is adopted, along with the method of Matricardi (2005) in order to compute the vertically integrated COD for a number of subcolumns within each model column. From these subcolumn values a mean COD for the cloudy part of the column is derived. A total COD is then computed as the average of the column mean COD over 5 × 5 columns centred around the column containing the relevant ground station.

*P22 onward, Tables reporting biases: (1) can the authors add to the captions the information on the asterisk purpose, so that the reader does not have to look through the text? (2) Would it be informative to report also these values expressed in %?*

Our response: This is now corrected for in the text, and both RMSE and MBE are expressed in %.

Changes to the manuscript:

The Table caption was updated to explain the asterisk, and all values were also normalised with the mean of the reference irradiance or optical depth, in order to give a relative MBE and RMSE.

*Spotted typos*
*P14 L 329: lookup table*

*P14 L 341: \citep instead of \cite for Crnivec and Mayer 2019*

Our response: these two typos are corrected in the text.

Changes to the manuscript:

- These two typos were corrected in the text.

Second referee (**https://doi.org/10.5194/amt-2022-335-RC2**)

*This is an interesting study that addresses a very relevant topic, namely the use of PV as a surrogate radiation measurement device to acquire a much larger spatial coverage than the current relatively sparse coverage of radiation measurement devices.*

*In my view, the work is very well done in terms of approach and analysis. My comments merely address the presentation and availability of software and data. These points should be easily addressed hence I have suggested minor revisions.*

*I have a couple of issues that I would like to point out. First concerning open sciences and the lack thereof in the review process. This maybe more a complaint to AMT than to the authors although I would also encourage them to embrace open science a bit more.*

1. *This review process cannot be done properly if the authors provide their data in a closed github repository, for which the anonymous reviewers need to reveal themselves in order to see the software. This goes against my definition of open science in my view, and in my view should be prohibited by the journal. I do not think this level of secrecy is constructive nor necessary.*
2. *In line with the previous comments. It would be nice if all software is made available open source, because from the paper it is not entirely clear what are datasets and what are tools that the readers can use themselves to reproduce or slightly alter the setups. Since the repository is closed, the reviewers cannot check.*

Our response: The software will be made open source once the manuscript is published. The decision to wait until publication was approved by the journal.

*Then concerning the manuscript:*

1. *I would like the authors to reflect more about the feasibility of this endeavour for operational use. They have tested this for a well-designed campaign / setup, but to me it is still a bit unclear whether the authors, based on their findings, expect this to be a realistic road to operational use. They are very careful in their comments in 588, and I would like to challenge them to be a bit more outspoken.*

Our response: The algorithm presented here is simply the first step in using PV data as an input for data assimilation in weather models. It is clear from the work that this is feasible, however as pointed out, moving towards operational use would require several further steps. The largest source of bias comes from the calibration step, in particular the effect of temperature. Access to direct current (DC) data on the inverter level would allow a much more accurate extraction of the irradiance, without the confounding effect of temperature, since the dependence of MPP current on temperature is an order of magnitude smaller than that of MPP voltage and power. Since inverter data is commonly available in industry, this should be possible provided one has the legal right to access the data. Additionally, the data pre-processing needs to be automated. For instance, one needs to exclude PV systems and/or data subsets that do not meet certain criteria such as shading of PV modules or inverter clipping. This could be done by developing a hybrid approach using both physical modelling and artificial intelligence (AI) algorithms for pattern recognition. In addition, the calibration procedure itself could be augmented with AI, to find the unknown parameters more effectively. Once these aspects are overcome and appropriate agreements with industry partners are made for access to the data, there should be nothing standing in the way of operational use.

Changes to the manuscript:

- The following was added to the manuscript

The ultimate goal of this work is to show that PV power data can be used to infer global horizontal irradiance and optical properties of the atmosphere, and the algorithm presented here is the first step in this direction. Although it is clear from the above that this is feasible, moving towards operational use would require several further steps. The largest source of bias comes from the calibration step, in particular the effect of temperature. Access to direct current (DC) data on the inverter level would allow a much more accurate extraction of the irradiance, without the confounding effect of temperature, since the dependence of MPP current on temperature is an order of magnitude smaller than that of MPP voltage and power. Since inverter data is commonly available in industry, this should be possible provided one has the legal right to access the data. Additionally, the data pre-processing needs to be automated. For instance, one needs to exclude PV systems and/or data subsets that do not meet certain criteria such as shading of PV modules or inverter clipping, and one needs to be able to detect clear sky periods automatically, even if the system orientation is unknown. This could be done by developing a hybrid approach using both physical modelling and artificial intelligence (AI) algorithms for pattern recognition. In addition, the calibration procedure itself could be augmented with AI, to find the unknown parameters more effectively.

Once these aspects are overcome and appropriate agreements with industry partners are made for access to the data, there should be nothing standing in the way of operational use. If this could be achieved at a large scale it would allow a better characterisation of solar irradiance at the ground and open up several possibilities for improving PV power forecasts at different time horizons. At shorter time (sub-hourly) scales one could use the additional information on irradiance variability as further input to empirical forecasts based on statistical methods, whereas at longer time scales (more than three hours) these data could be assimilated into weather models. In order for this to make a difference one needs a much larger data set of PV systems for the analysis, which then requires further automation. First steps in this direction are currently being explored.

2. *Table 9. I would like the authors to comment a bit more on the restrictions presented in the _limits_ column. Are these very restrictive? How much does this influence potential operational use in the future?*

Our response: This is discussed in more detail in the response to referee 1. As shown in Figure 2 in the preprint (this figure is updated to make it more explicit, and the label for DISORT is changed to 1D LUT), the algorithm uses different lookup tables depending on the prevailing conditions. For more stable weather conditions such as clear or cloudy skies, the DISORT-based LUT is used for AOD or COD, whereas for broken cloud conditions the MYSTIC LUT is used, but in this case only to convert tilted to horizontal irradiance. The limits for the MYSTIC LUT were also chosen to reduce computational time and could be extended slightly, with regards the SZA and tilt angles. For operational use, the data would be separated based on cloud fraction, and one would decide whether to use either the DISORT or MYSTIC LUTs on a case-by-case basis. This would then dictate exactly which limits are chosen for the MYSTIC 3D LUT.

Changes to the manuscript: see comments to referee 1.

3. *I find it a little worrying that the authors base their weather model data on a model that according to their own comments went into retiring in 2021. Why not base the results on ICON? If this ever should become a method for operational use it makes a lot more sense to compare it against the local state of the art.*

Our response: That is correct, however the data made available by the DWD for validation is from the COSMO model, since at the time of the measurement campaigns this model was the state of the art at the German weather centre.

Changes to the manuscript:

- The following was added:

"Since the measurement campaigns took place in 2018 and 2019, …" was added in L 385.

  4. *Eq. 10: what is the origin of this equation and what is the physics behind it?*

Our response: The liquid water content (LWC) of a cloud describes the mass of water droplets per unit volume, and the liquid water path (LWP) is the integral of the LWC over the height of the cloud, i.e. LWP = LWC·h (1) with h being the vertical extension of the cloud. The LWC can be calculated from the integral of all cloud droplets (n(r)), their volume ($4/3·\pi·r^3$) and the density of water ρ, i.e.

LWC = ∫ n(r) ·$\rho_{H2O}$ ·$4/3·\pi·r^3$·dr (2).

For large Mie extinction ($2·\pi·r/\lambda \gg 1$, λ being the wavelength), which is justified since clouds appear (mostly) to be white in the solar spectrum, the volume extinction coefficient ($Q_{ext}$) is the integral $Q_{ext}$ over all radii, i.e. $Q_{ext}$ = ∫ n(r)·$2·\pi·r^2$ ·dr (3), with the cloud optical depth being $\tau_c$ = $Q_{ext}$ ·h (4).

Using the definition of the effective radius (re) re = [∫$\pi·r^3$·dr]/[∫$\pi·r^2$·dr] (5), inserting all equations into (1) and rearranging you obtain equation (10) in the text.

Changes to the manuscript:

- In line 281, the following was added:

The derivation of this equation [see for instance Petty, (2006)] assumes large Mie extinction, which is justified since clouds appear to be (mostly) white in the solar spectrum.

  5. *Eq. 12: how sensitive are results to the chosen thresholds?*

Our response: The cloud mask is designed in a way to create a series of binary states, cloudy or clear. The calculated cloud fraction depends less on the exact threshold used (in our case 0.8), but more on the window size chosen for the moving average. The algorithm calculates a cloud fraction by averaging the binary cloud mask, and comparison with concurrent cloud camera retrievals showed that 60 minutes was a reasonable averaging time to use, when averaging a cloud mask create at one minute resolution. However, the algorithm is limited by the viewing angle of the respective PV system, so it can be inaccurate when there are many clouds on the horizon, for instance.

Changes to the manuscript:

- The following was added to line 313 below Eq. 12

Varying the thresholds in Eq. (12) shows that the cloud fraction computed in this way depends less on the exact threshold used, but more on the window size chosen for the moving average. Indeed, comparison with concurrent cloud camera retrievals shows that 60 minutes is a reasonable averaging time to use, when averaging a cloud mask created with data at one minute resolution. However, the algorithm is limited by the viewing angle of the respective PV system or pyranometer, so it can be inaccurate when there are many clouds on the horizon, for instance.

  6. *Line 344. I do not understand how a PV setup sees the whole sky. Please clarify.*

Our response: We decided to end this sentence after "cloud fraction.", otherwise we agree that it is confusing.

Changes to the manuscript:

- Removed the last part of this sentence, after "cloud fraction".

7. *Figure 8: The 15-min averages have an almost twice as large bias compared to the 1-min data. Is this because the added measurements are so much worse?*

Our response: As discussed above in the response to the first referee, this is due to poor performance of the calibration algorithm in this case. If the calibration algorithm performs poorly, large biases can be expected. It has also been checked whether the presence of erroneous data could lead to this, however this affects the RMSE more than the bias (as discussed in the first response).

Changes to manuscript:

- These changes were discussed in response to referee 1.

*If these points are addressed, I think the authors provide a nice basis for future work on this topic. I am glad that their results are based on physics.*

**References**

Becker, R. and Behrens, K.: Quality assessment of heterogeneous surface radiation network data, Adv. Sci. Res., 8, 93–97, https://doi.org/10.5194/asr-8-93-2012, 2012.

Elsinga, B., van Sark, W., and Ramaekers, L.: Inverse photovoltaic yield model for global horizontal irradiance reconstruction, Energy Sci Eng, 5, 226–239, https://doi.org/10.1002/ese3.162, 2017.

Engerer, N. A. and Mills, F. P.: KPV: A clear-sky index for photovoltaics, Solar Energy, 105, 679–693, https://doi.org/10.1016/J.SOLENER.2014.04.019, 2014.

Halilovic, S., Bright, J. M., Herzberg, W., and Killinger, S.: An analytical approach for estimating the global horizontal from the global tilted irradiance, Solar Energy, 188, 1042–1053, https://doi.org/10.1016/J.SOLENER.2019.06.027, 2019.

Hess, M., Koepke, P., and Schult, I.: Optical properties of aerosols and clouds, Bull. Amer. Meteor. Soc., 79, 831–844, https://doi.org/10.1175/1520-0477(1998)079<0831:OPOAAC>2.0.CO;2, 1998.

Killinger, S., Braam, F., Müller, B., Wille-Haussmann, B., and McKenna, R.: Projection of power generation between differently-oriented PV systems, Solar Energy, 136, 153–165, https://doi.org/10.1016/J.SOLENER.2016.06.075, 2016.

Mandija, F., Chavez-Perez, V. M., Nieto, R., Sicard, M., Danylevsky, V., Añel, J. A., and Gimeno, L.: The climatology of dust events over the European continent using data of the BSC-DREAM8b model, Atmos Res, 209, 144–162, https://doi.org/https://doi.org/10.1016/j.atmosres.2018.03.006, 2018.

Matricardi, M.: The inclusion of aerosols and clouds in RTIASI, the ECMWF fast radiative transfer model for the infrared atmospheric sounding interferometer., https://doi.org/10.21957/1krvb28ql, 2005.

McArthur, L. J. B.: Baseline Surface Radiation Network (BSRN). Operations Manual Version 2.1, Downsview, Ontario, 2005.

Petty, G. W.: A First Course in Atmospheric Radiation, Second., Sundog Publishing, Madison, Wisconsin, 452 pp., 2006.

Sauer, D. U.: Untersuchungen zum Einsatz und Entwicklung von Simulationsmodellen für die Auslegung von Photovoltaik-Systemen, Diplomarbeit, Technische Hochschule Darmstadt, 1–247 pp., https://doi.org/10.13140/RG.2.1.1833.7366, 1994.

Scheck, L., Weissmann, M., and Mayer, B.: Efficient Methods to Account for Cloud-Top Inclination and Cloud Overlap in Synthetic Visible Satellite Images Efficient Methods to Account for Cloud-Top Inclination and Cloud Overlap in Synthetic Visible Satellite Images, 35, 665–685, https://doi.org/10.1175/JTECH-D-17-0057.1, 2018.